# *Con4m*: Context-aware Consistency Learning Framework for Segmented Time Series Classification

**Junru Chen**
Zhejiang University
jrchen_cali@zju.edu.cn

**Tianyu Cao**
Zhejiang University
ty.cao@zju.edu.cn

**Jing Xu**
State Grid Power Supply Co. Ltd.
ltxu1111@gmail.com

**Jiahe Li**
Zhejiang University
jiaheli@zju.edu.cn

**Zhilong Chen**
Zhejiang University
zhilongchen@zju.edu.cn

**Tao Xiao**
State Grid Power Supply Co. Ltd.
xtxjtu@163.com

**Yang Yang**[†]
Zhejiang University
yangya@zju.edu.cn

## Abstract

Time Series Classification (TSC) encompasses two settings: classifying entire sequences or classifying segmented subsequences. The raw time series for segmented TSC usually contain **M**ultiple classes with **V**arying **D**uration of each class (*MVD*). Therefore, the characteristics of *MVD* pose unique challenges for segmented TSC, yet have been largely overlooked by existing works. Specifically, there exists a natural temporal dependency between consecutive instances (segments) to be classified within *MVD*. However, mainstream TSC models rely on the assumption of independent and identically distributed (*i.i.d.*), focusing on independently modeling each segment. Additionally, annotators with varying expertise may provide inconsistent boundary labels, leading to unstable performance of noise-free TSC models. To address these challenges, we first formally demonstrate that valuable contextual information enhances the discriminative power of classification instances. Leveraging the contextual priors of *MVD* at both the data and label levels, we propose a novel consistency learning framework *Con4m*, which effectively utilizes contextual information more conducive to discriminating consecutive segments in segmented TSC tasks, while harmonizing inconsistent boundary labels for training. Extensive experiments across multiple datasets validate the effectiveness of *Con4m* in handling segmented TSC tasks on *MVD*. The source code is available at https://github.com/MrNobodyCali/Con4m.

## 1 Introduction

Time Series Classification (TSC) is one of the most challenging problems in the field of machine learning. TSC aims to assign labels to a series of temporally ordered data points. These points either form a complete sequence or are subsequences (segments) resulting from the segmentation of a long time series. Existing works [50, 20] largely focus on the assumption of independent and identically distributed (*i.i.d.*), in which case each sequence or segment is regarded as an independent instance to be classified, not differentiating between these two settings. In fact, for many practical applications, the raw time series before segmentation for segmented TSC tasks contain **M**ultiple classes with

---

[†] Corresponding author.

38th Conference on Neural Information Processing Systems (NeurIPS 2024).

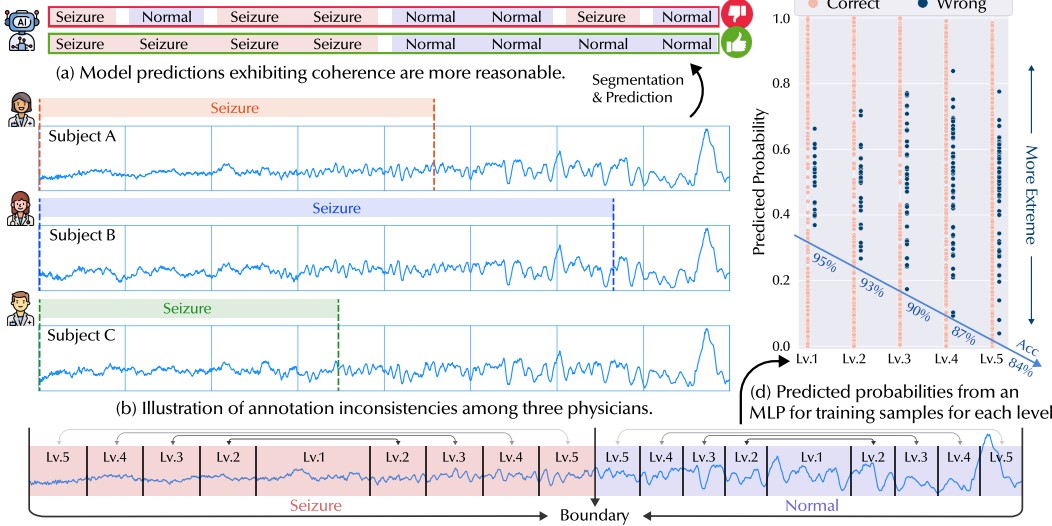

(a) Model predictions exhibiting coherence are more reasonable.

(b) Illustration of annotation inconsistencies among three physicians.

(c) Schematic diagram of the division of each class sequence in (d), with higher-level data closer to the boundary.

(d) Predicted probabilities from an MLP for training samples for each level.

Figure 1: (a) Reasonable model predictions exhibit coherence across consecutive segments rather than repeated interruptions. (b) In the healthcare domain, different physicians have varying annotations regarding the start and end times of seizure waves. (c) Based on the proximity to the boundary, we divide each class sequence into 5 levels, from which an equal number of segments are sampled. A one-layer MLP is trained on the segments from each level respectively for the same number of epochs. (d) We visualize the predicted probability of the trained MLP for each level. We observe that as the segments approach the boundaries, the model finds it increasingly challenging to make correct classifications, resulting in more extreme wrong predictions. This strongly underscores the significance of handling boundary segments.

**Varying Duration of each class (*MVD*).** For example, in the healthcare domain, the brain signals of epileptic patients often record over several days, encompassing multiple seizure onsets, each with varying durations and intervals. In the field of activity recognition, sensors continuously record users' behavior data, including walking, riding, and running, among other activities, each with varying durations. Therefore, the characteristics of the raw *MVD* lead to the uniqueness of segmented TSC tasks. Given this, our research concentrates on effectively modeling segmented TSC tasks based on *MVD*, presenting distinctive challenges.

**(1) Leveraging contextual information.** In contrast to TSC tasks for complete sequences, in which classified sequences are relatively independent, there exist natural temporal dependencies between consecutive classified segments for segmented TSC. We take the seizure detection task as an example, in which given the brain signals of epileptic patients, the model should identify whether a segment includes seizure waves or not. As illustrated in Figure 1(a), given the sustained nature of seizure onsets, the model predictions for consecutive segments should exhibit coherence, with seizure and normal predictions appearing in continuous and concentrated patterns. However, mainstream TSC models [58, 71, 61] focus on the *i.i.d.* assumption and model the internal context within each segment to be classified, largely overlooking the dependencies between consecutive segments.

In the domain of video analysis, works in temporal action segmentation (TAS) [17] have modeled the temporal dependency between different video frames and made frame-wise predictions. However, unlike the I3D features [8] used as input in these works, time series lack a unified pretrained model for feature extraction, and the dependency between segments is more variable and ambiguous. Furthermore, TAS works focus on modeling the dependency of instances from a data perspective, without explicitly leveraging contextual label information. Therefore, how to leverage contextual information in segmented TSC tasks to make more reasonable classifications is crucial and challenging.

**(2) Inconsistent boundary labels.** In domains with precious labels, different annotators collaboratively contribute annotations. The raw annotations of *MVD* typically include the start and end times for each class. However, due to inherent ambiguity and a lack of unified quantification standards, for *MVD*, the boundaries between states are not clearly defined, or the transitional state itself represent a mixed state. Consequently, behind inconsistent labels, there is no artificially defined true label. Therefore, our work aims to harmonize this inconsistency as much as possible to reduce the instability

of model training and enhance its performance. Returning to Figure 1(b), in the seizure detection task, owing to the natural fuzzy transition from seizure onset to completely normal, different physicians have varying experiences regarding when seizure waves terminate.

Furthermore, inconsistent boundary labeling causes that boundary segments with similar patterns may have opposite labels, leading to unstable model training. To validate the detrimental impact, as shown in Figure 1(c), we divide each class sequence in the seizure detection task into 5 levels, where higher levels indicate proximity to the boundary. We then sample an equal number of balanced binary segments for each level. Subsequently, a one-layer MLP is trained for the same number of epochs on the segments from each level respectively. Figure 1(d) visualizes the results after training. We observe that as the level increases (closer to the boundary), the model's accuracy steadily decreases, and erroneous predictions become more extreme. The results highlight the significant detrimental impact of inconsistent boundary labels on noise-free model performance.

Noisy label learning (NLL) [56] aims to learn robust models from data containing corrupted labels. While the inconsistent labels in *MVD* are not intentionally corrupted but rather stem from implicit discrepancies due to experiential differences, NLL remains the most relevant approach to address such discrepancies. To the best of our knowledge, Scale-teaching [47] and SREA [9] are the only NLL works specifically designed for time series and are thus the most relevant to our work. However, they also face the issue of overlooking contextual dependencies across consecutive time segments, posing the challenge of handling inconsistent boundary labels using context during training.

To overcome the challenges above, we propose *Con4m*–a label **Con**sistency learning framework, which leverages effective **Con**textual information, achieving **Co**herent predictions and **Con**tinuous representations for seg**m**ented TSC tasks, while harmonizing inconsistent boundary labels for training. Specifically, we first formally demonstrate that valuable contextual information enhances the discriminative power of classification instances. Based on the insights, by incorporating prior knowledge of data locality and label coherence, we guide and constrain the model to focus on contextual information more conducive to discriminating consecutive segments in segmented TSC tasks. Meanwhile, leveraging model predictions that thoroughly encompass contextual information, *Con4m* progressively changes the training labels in an adaptive manner to harmonize inconsistent labels across consecutive segments. This leads to a more robust model. Our contributions are summarized as follows:

**(1)** We are the first to propose a practical consistency learning framework *Con4m* for the segmented TSC based on the raw *MVD*. **(2)** By comprehensively integrating prior knowledge from the data and label perspectives, we guide the model to focus on effective contextual information. Based on context-aware predictions, a progressive harmonization approach for handling inconsistent training labels is designed to yield a more robust model. **(3)** Extensive experiments on three public and one private *MVD* datasets demonstrate the superior performance of *Con4m*. The *Con4m*'s ability to harmonize inconsistent labels is further verified by the label substitution experiment and case study.

## 2 Theoretical Analysis

In this section, we aim to formally demonstrate the benefit of contextual information for classification tasks, and to establish the existence of an upper bound for this benefit. Consequently, by introducing prior knowledge, we can guide the model to focus on valuable contextual information more conducive to improving the benefit for segmented TSC tasks.

Assuming that the random variables of the instances to be classified and the corresponding labels are denoted as $\mathrm{x}_t$ and $\mathrm{y}_t$. $\mathbb{A}_t$ represents the contextual instance set introduced for $\mathrm{x}_t$. $x_{\mathbb{A}_t}$ denotes the random variable for the contextual instance set. Mutual information measures the correlation between two random variables. In a classification task, a higher correlation between instances and labels indicates that the instances are more easily distinguishable by the labels. This benefits the classification task, making it more readily addressable. Therefore, from an information-theoretic perspective, we elucidate the benefit of contextual information through the following theorem.

**Theorem 2.1.** *The more the introduced contextual instance set enhance the discriminative power of the target instance, the greater the benefit for the classification task.*

*Proof.* Firstly, we establish that the introduction of contextual information does not compromise classification tasks, *i.e.*, it does not diminish the correlation between instances and labels.

$$\mathbb{I}(\mathrm{y}_t; \mathrm{x}_t, \mathrm{x}_{\mathbb{A}_t}) = \mathbb{I}(\mathrm{y}_t; \mathrm{x}_{\mathbb{A}_t}|\mathrm{x}_t) + \mathbb{I}(\mathrm{y}_t; \mathrm{x}_t) \geq \mathbb{I}(\mathrm{y}_t; \mathrm{x}_t). \tag{1}$$

The inequality holds due to the non-negativity of conditional mutual information.

According to (1), the increase in $\mathbb{I}(y_t; x_{\mathbb{A}_t}|x_t)$ determines the extent to which the introduction of contextual information can be beneficial for classification tasks. Expanding $\mathbb{I}(y_t; x_{\mathbb{A}_t}|x_t)$, we have:

$$
\begin{aligned}
\mathbb{I}(y_t; x_{\mathbb{A}_t}|x_t) &= \sum_{x_t} p(x_t) \sum_{x_{\mathbb{A}_t}} \sum_{y_t} p(y_t, x_{\mathbb{A}_t}|x_t) \log \frac{p(y_t, x_{\mathbb{A}_t}|x_t)}{p(y_t|x_t)p(x_{\mathbb{A}_t}|x_t)} \\
&= \sum_{x_t} p(x_t) \sum_{x_{\mathbb{A}_t}} \sum_{y_t} p(y_t|x_t, x_{\mathbb{A}_t})p(x_{\mathbb{A}_t}|x_t) \log \frac{p(y_t|x_t, x_{\mathbb{A}_t})}{p(y_t|x_t)} \\
&= \sum_{x_t} p(x_t) \sum_{x_{\mathbb{A}_t}} p(x_{\mathbb{A}_t}|x_t) D_{\mathrm{KL}}(p(y_t|x_t, x_{\mathbb{A}_t})\|p(y_t|x_t)).
\end{aligned}
$$

Given a fixed instance $x_t$ and the inherent distribution $p(y_t|x_t)$ of the data, the KL divergence is a convex function for $x_{\mathbb{A}_t}$ that attains its minimum at $p(y_t|x_t, x_{\mathbb{A}_t}) = p(y_t|x_t)$. As $p(y_t|x_t, x_{\mathbb{A}_t})$ approaches the boundary of the probability space, where the predictive probability of one class approaches 1 and the rest approach 0, the value of KL divergence increases. A stronger discriminative power regarding $x_t$ implies less uncertainty regarding $y_t$, which is equivalent to approaching the boundary of the probability space.

Due to the convexity of the KL divergence and the boundedness of $p(y_t|x_t, x_{\mathbb{A}_t})$, there exists a contextual instance set in the data that maximizes $D_{\mathrm{KL}}(p(y_t|x_t, x_{\mathbb{A}_t})\|p(y_t|x_t))$. We denote the instance set as $\mathbb{A}_t^*$ and the maximum value of KL divergence as $D_t^*$. Besides, we note that $\sum_{x_{\mathbb{A}_t}} p(x_{\mathbb{A}_t}|x_t) = 1$. Hence, we can obtain the upper bound for the information gain $\mathbb{I}(y_t; x_{\mathbb{A}_t}|x_t) \leq \sum_{x_t} p(x_t) \sum_{x_{\mathbb{A}_t}} p(x_{\mathbb{A}_t}|x_t)D_t^* \leq \sum_{x_t} p(x_t)D_t^*$. The convexity of the KL divergence also implies monotonicity, indicating that as $A_t$ approaches $A_t^*$, the KL divergence increases, leading to a greater information gain for the classification task. □

According to Theorem 2.1, valuable contextual information enhances the discriminative power of the instances. While the optimal instance set $A_t^*$ is challenging to directly obtain or optimize, focusing the model on contextual instances more likely to be included in $A_t^*$ is beneficial for enhancing the performance of the classification task. Furthermore, $x_{\mathbb{A}_t}$ not only contains information at the data level but also encompasses information at the label level (which can be replaced with $y_{\mathbb{A}_t}$). Therefore, we can guide the model to focus on contextual information more conducive to segmented TSC tasks by simultaneously introducing prior knowledge from both the data and label perspectives.

## 3  The *Con4m* Method

In this section, we introduce the details of *Con4m*. Based on the insights of Theorem 2.1, we introduce contextual prior knowledge of data locality (Sec. 3.1) and label coherence (Sec. 3.2) to guide the model to focus on contextual information more conducive to discriminating consecutive segments in segmented TSC tasks. In Sec. 3.3, inspired by the idea of noisy label learning, we propose a label harmonization framework to achieve a more robust model. Before delving into the details of *Con4m*, we provide the formal definition of the segmented TSC task in our work.

**Definition 3.1.** *Given a time interval comprising of $T$ consecutive time points and labels, denoted as $(X, Y) = \{(X_1, Y_1), (X_2, Y_2), \ldots, (X_T, Y_T)\}$, a $w$-length sliding window with stride length $r$ is employed for segmentation. $(X, Y)$ is partitioned into $L$ time segments, represented as $(x, y) = \{(x_i, y_i) = (\{X_{(i-1) \times r+1}, \ldots, X_{(i-1) \times r+w}\}, Majority(\{Y_{(i-1) \times r+1}, \ldots, Y_{(i-1) \times r+w}\}))|i = 1, \ldots, L\}$. The model is tasked with predicting segmented labels $y_i$ for each time segment $x_i$.*

### 3.1  Continuous Contextual Representation Encoder

Local continuity is an inherent attribute of *MVD*, meaning each class should be locally continuous and only change at its actual boundary. Smoothing with a Gaussian kernel [18, 16, 66] promotes the continuity of representations of time segments in a local temporal window. This not only helps the model make similar predictions of consecutive segments within the same class but also aligns with the gradual nature of class transitions. Furthermore, for graph neural networks based on the

homophily assumption, aggregating neighbor information belonging to the same class can improve the discriminative power of the target instance [49, 73]. Therefore, we introduce the Gaussian prior to guide the model to focus on contextual instances $\mathbb{A}_t$ proximate to the target instance.

Vanilla self-attention [15] with point-wise attention computations often fail to obtain continuous representations after aggregation. Therefore, we use the Gaussian kernel $\Phi(x, y|\sigma)$ as prior weights to aggregate neighbors to obtain smoother representations. Since the neighbors of boundary segments may belong to different classes, we allow each segment to learn its own scale parameter $\sigma$. Formally, as Figure 2(a) shows, the two-branch **Con-Attention** in the $l$-th layer is:

$$Q, K, V_s, V_g, \sigma = c^{l-1}W_Q^l, c^{l-1}W_K^l, c^{l-1}W_{V_s}^l, c^{l-1}W_{V_g}^l, c^{l-1}W_\sigma^l,$$

$$S^l = \text{SoftMax}\left(\frac{QK^\top}{\sqrt{d}}\right), \quad G^l = \text{Rescale}\left(\left[\frac{1}{\sqrt{2\pi}\sigma_i}\exp\left(-\frac{|j-i|^2}{2\sigma_i^2}\right)\right]_{i,j\in\{1,...,L\}}\right),$$

$$z_s^l = S^l V_s, \quad z_g^l = G^l V_g, \quad z^l = \text{Fusion}(z_s^l, z_g^l),$$

where $L$ is the number of consecutive segments, $d$ is the dimension of hidden representations, $c^{l-1} \in \mathbb{R}^{L\times d}$ is the output representations of the $l-1$-th layer, and $W_*^l \in \mathbb{R}^{d\times d}$ are all learnable matrices. Rescale($\cdot$) refers to row normalization by index $i$. To distinguish between two computational branches, we use $g/G$ to represent the branch based on Gaussian prior, and $s/S$ to represent the branch based on self-attention. $S^l$ and $G^l$ are the aggregation weights. We use the conventional attention mechanism [4] to adaptively fuse $z_s^l$ and $z_g^l$. Finally, as illustrated in Figure 2(a), by stacking the multi-head version of Con-Attention layers, we construct Con-Transformer, which serves as the backbone of the continuous encoder of *Con4m* to obtain final representations $c$. We employ learnable absolute positional encoding (APE) [23] for the input representations.

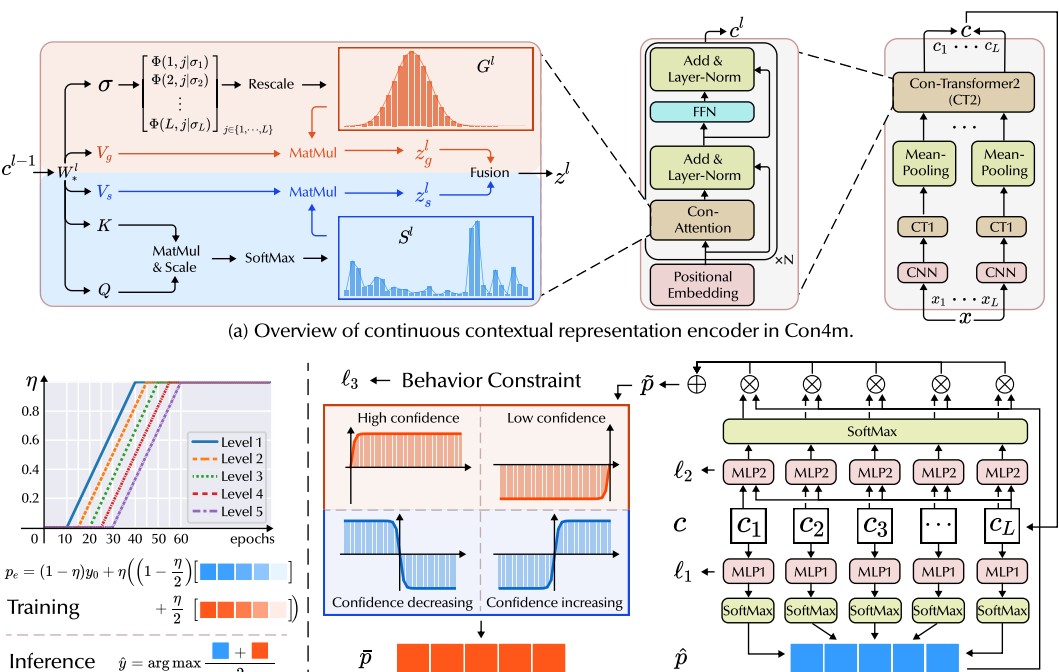

(a) Overview of continuous contextual representation encoder in Con4m.

(b) Overview of context-aware coherent class prediction and consistent label training framework in Con4m.

Figure 2: Overview of *Con4m*. (a) Overview of continuous contextual representation encoder in *Con4m*. The leftmost part shows the details of Con-Attention. The right part of the figure shows the architecture of Con-Transformer and the whole encoder of *Con4m*. (b) Overview of context-aware coherent class prediction and consistent label training framework in *Con4m*. The right part describes the neighbor class consistency discrimination task and the prediction behavior constraint. The leftmost part presents the training and inference details for label harmonization.

## 3.2 Context-aware Coherent Class Prediction

In the segmented TSC task of *MVD*, consecutive time segments not only provide contextual information at the data level but also possess their own class information. As depicted in Figure 1(a), considering the persistence of each class and the gradual nature of class transitions, the model's predictions should exhibit more coherence and concentration, rather than being interspersed. Therefore, we integrate and constrain the model's predictions from both the individual and holistic perspectives to achieve more coherent predictions.

**Neighbor Class Consistency Discrimination.** In graphs, label propagation algorithms [30, 33] are often utilized to refine and smooth the predictions of neighbor instances, thereby enhancing their discrimination. Drawing inspiration from this, by weightedly aggregating predictions from similar time segments, the model can focus on contexts $\mathbb{A}_t$ more likely to belong to the same class as the target segment. Although there is no explicit graph structure between time segments, we can train a discriminator to determine whether two segments belong to the same class. The model then aggregates the contextual class predictions based on the discriminator's outputs, thus making more robust predictions. As the right part of Figure 2(b) shows, we formalize this process as follows:

$$\hat{R} = \text{SoftMax}\left(\left[\text{MLP}_2\left(c_i\|c_j\right)\right]_{i,j\in\{1,\dots,L\}}\right), \quad \hat{p} = \text{SoftMax}\left(\text{MLP}_1\left(c\right)\right), \quad \tilde{p} = \hat{R}_{:,:,1}\hat{p},$$

where $\hat{R} \in \mathbb{R}^{L\times L\times 2}$ is the probability of whether two segments in the same time interval belong to the same class and $(\cdot\|\cdot)$ denotes tensor concatenation. $\hat{p}$ represents the model's independent prediction for a segment, while $\tilde{p}$ denotes the context-aware prediction that incorporates the results from neighboring segments. We then define the two training losses as $\ell_1 = \text{CrossEntropy}(\hat{p}, y)$ and $\ell_2 = \text{CrossEntropy}(\hat{R}, \tilde{Y})$, where $\tilde{Y} = [\mathbf{1}_{y_i=y_j}]_{i,j\in\{1,\dots,L\}}$. Given that $\ell_1$ and $\ell_2$ are of the same magnitude, we equally sum them as the final loss.

**Prediction Behavior Constraint.** Unlike graphs, there exists a holistic temporal relationship between consecutive time segments. Therefore, we should further constrain the overall predictive behavior along the time axis. For *MVD*, as Figure 1(a) shows, within a suitably chosen time interval, consecutive segments almost span at most two classes. Therefore, we ensure the monotonicity of predictions across consecutive segments through hard constraints, thereby utilizing contextual label information $y_{\mathbb{A}_t}$ to integrate and refine predictions across these segments.

As shown in the middle part of Figure 2(b), for each class in the predictions, there are only four prediction behaviors for consecutive segments, namely *high confidence*, *low confidence*, *confidence decreasing*, and *confidence increasing*. To constrain the behavior, we use function fitting to integrate $\tilde{p}$. Considering the wide applicability, we opt for the hyperbolic tangent function (*i.e.*, Tanh) as our basis. Formally, we introduce four tunable parameters to exactly fit the monotonicity as:

$$\bar{p} = \text{Tanh}(x|a, k, b, h) = a \times \text{Tanh}\left(k \times (x + b)\right) + h,$$

where parameter $a$ constrains the range of the function's values, $k$ controls the slope of the transition of the function, $b$ and $h$ adjust the symmetry center of the function, and $x$ is the given free vector in the x-coordinate. We use the MSE loss to fit the contextual predictions $\tilde{p}$ as $\ell_3 = \|\text{Tanh}(x|a, k, b, h) - \tilde{p}\|^2$. It deserves to be emphasized that $\tilde{p}$ in the process has no gradient and therefore does not affect the parameters of the encoder. Please see Appendix B for more fitting details.

After function fitting, we obtain independent predictions $\hat{p}$ for each segment and constrained predictions $\bar{p}$ that leverage contextual label information. For the inference stage, we use the average of them as the final coherent predictions, *i.e.*, $\hat{y} = \arg\max (\hat{p} + \bar{p})/2$.

## 3.3 Label Consistency Training Framework

Due to inherent ambiguity, the annotation of *MVD* often lacks quantitative criteria, resulting in experiential differences across individuals. Such discrepancies are detrimental to models and we propose a training framework to enable *Con4m* to adaptively harmonize inconsistent labels.

**Learning from easy to hard.** We are based on the fact that although people may have differences in the fuzzy transitions between classes, they tend to reach an agreement on the most significant core part of each class. In other words, the empirical differences become more apparent when approaching the transitions. Therefore, we adopt curriculum learning techniques to help the model learn instances

from the easy (core) to the hard (transition) part. Formally (see the diagram in Figure 1(b)), for a continuous $K$-length class, we divide it into $N_l = 5$ equally sized levels as follows:

$$\left( \lceil (N_l - 1)\frac{K}{2N_l} \rceil, \lfloor (N_l + 1)\frac{K}{2N_l} \rfloor \right); \cdots ; \left[ 1, \lceil \frac{K}{2N_l} \rceil \right) \bigcup \left( \lfloor (2N_l - 1)\frac{K}{2N_l} \rfloor, K \right]. \quad (2)$$

Then we sample the same number of time intervals from each level. The higher the level, the more apparent the inconsistency. Therefore, as the left part of Figure 2(b) shows, during the training stage, *Con4m* learns the time intervals in order from low to high levels, with a lag gap of $E_g = 5$ epochs.

**Harmonizing inconsistent labels.** Inspired by the idea of noisy label learning, we gradually change the raw labels to harmonize the inconsistency. The model preferentially changes the labels of the core segments that are easier to reach a consensus, which can avoid overfitting of uncertain labels. Moreover, the model will consider both the independent and constrained predictions to robustly change inconsistent labels. Specifically, given the initial label $y_0$, we update the labels $y_e = \arg\max p_e$ for the $e$-th epoch, where $p_e$ is obtained as follows:

$$\hat{p}_e^5 = \omega_e \cdot [\hat{p}_{e-m}]_{m \in \{0,...,4\}}, \quad \bar{p}_e^5 = \omega_e \cdot [\bar{p}_{e-m}]_{m \in \{0,...,4\}},$$

$$p_e = (1 - \eta) y_0 + \eta \left( \left(1 - \frac{\eta}{2}\right) \hat{p}_e^5 + \frac{\eta}{2} \bar{p}_e^5 \right),$$

where $\omega_e = \text{Rescale}([\exp((e - m)/2)]_{m \in \{0,...,4\}})$ is the exponentially averaged weight vector to aggregate the predictions of the latest 5 epochs to achieve a more robust label update. $\hat{p}_{e-m}$ and $\bar{p}_{e-m}$ are the independent and constrained predictions in the $e - m$-th epoch respectively and $\cdot$ denotes the dot product. The dynamic weighting factor, $\eta$, is used to adjust the degree of label update. As the left part of Figure 2(b) shows, $\eta$ linearly increases from 0 to 1 with $E_\eta$ epochs, gradually weakening the influence of the original labels. Besides, in the initial training stage, the model tends to improve independent predictions. As the accuracy of independent predictions increases, the model assigns a greater weight to the constrained predictions. See the hyperparameter analysis for $E_\eta$ in Appendix C.

## 4 Experiment

### 4.1 Experimental Setup

**Datasets.** In this work, we use three public [31, 7, 37] and one private *MVD* data to measure the performance of models. Specifically, the Tufts fNIRS to Mental Workload [31] data (**fNIRS**) contains brain activity recordings from adult humans performing controlled cognitive workload tasks. The **HHAR** (Heterogeneity Human Activity Recognition) dataset [7] captures sensor data from multiple smart devices to explore the impact of device heterogeneity on human activity recognition. The SleepEDF [37] data (**Sleep**) contains PolySomnoGraphic sleep records for subjects over a whole night. The private **SEEG** data records brain signals indicative of suspected pathological tissue within the brain of epileptic patients. More detailed descriptions can be found in Table 1 and Appendix D.

Table 1: Overview of *MVD* datasets used in this work.

| Data | Sample Frequency | # of Features | # of Classes | Subjects | Groups | Cross Validation | Total Intervals | Interval Length | Window Length | Slide Length | Total Segments |
|------|------|------|------|------|------|------|------|------|------|------|------|
| fNIRS | 5.2Hz | 8 | 2 | 68 | 4 | 12 | 4,080 | 38.46s | 4.81s | 0.96s | 146,880 |
| HHAR | 50Hz | 6 | 6 | 9 | 3 | 6 | 5,400 | 60s | 4s | 2s | 156,600 |
| Sleep | 100Hz | 2 | 5 | 154 | 3 | 6 | 6,000 | 40s | 2.5s | 1.25s | 186,000 |
| SEEG | 250Hz | 1 | 2 | 8 | 4 | 3 | 8,000 | 16s | 1s | 0.5s | 248,000 |

**Label disturbance.** We introduce a novel disturbance method to the raw labels $Y$ of the public datasets to simulate scenarios where labels are inconsistent. Specifically, we first look for the boundary points between different classes in a complete long *MVD* data. Then, we randomly determine with a 0.5 probability whether each boundary point should move forward or backward. Finally, we randomly select a new boundary point position from $r\%$ of the length of the class in the direction of the boundary movement. In this way, we can interfere with the boundaries and simulate label inconsistency. Meanwhile, a larger value of $r\%$ indicates a higher degree of label inconsistency. For SEEG dataset, inconsistent labels already exist in the raw data and we do not disturb it.

**Baselines.** We compare *Con4m* with state-of-art models from various domains, including two noisy label learning (NLL) models for time series classification (TSC): SREA [9] and Scale-teaching [47]

(Scale-T), three image classification models with noisy labels: SIGUA [28], UNICON [36] and Sel-CL [43], three supervised TSC models: MiniRocket [13], TimesNet [64] and PatchTST [51], and three temporal action segmentation (TAS) models: MS-TCN2 [42], ASFormer [68] and DiffAct [45]. See more detailed descriptions of the baselines in Appendix E.

**Implementation details.** We use cross-validation [39] to evaluate the model's generalization ability by partitioning the subjects in the data into non-overlapping subsets for training and testing. As shown in Table 1, for fNIRS and SEEG, we divide the subjects into 4 groups and follow the 2 training-1 validation-1 testing (2-1-1) setting to conduct experiments. We divide the HHAR and Sleep datasets into 3 groups and follow the 1-1-1 experimental setting. Notice that SEEG data is derived from real clinical datasets and annotated by multiple experts, resulting in naturally inconsistent labels. We employ a voting mechanism which brings annotators together to collectively decide the boundaries to minimize discrepancies in test labels. Considering the high cost of this approach, we do not apply it to the training and validation sets. Therefore, we leave the test group aside and only change the training and validation groups to conduct cross-validation. Finally, we only report the mean values of cross-validation results in the main context. See more details and the full results in Appendix G.

## 4.2 Label Disturbance Experiment

The average results over all cross-validation experiments are presented in Table 2. Overall, *Con4m* outperforms almost all baselines across all datasets and all disturbance ratios.

Table 2: Comparison with baseline methods in the testing $F_1$ score (%) on three datasets. The **best results** are in bold and we underline the second best results. The *worst results* are denoted in italics.

| | $r\%$ / Model | fNIRS [31] | | | HHAR [7] | | | Sleep [37] | | | SEEG |
|---|---|---|---|---|---|---|---|---|---|---|---|
| | | 0% | 20% | 40% | 0% | 20% | 40% | 0% | 20% | 40% | raw |
| TAS | MS-TCN2 [42] | 71.48 | 70.99 | 69.40 | 69.79 | 66.72 | 62.29 | 60.07 | 59.03 | 56.17 | 61.88 |
| | ASFormer [68] | **71.69** | 70.75 | 69.18 | 62.52 | 60.92 | 60.77 | 59.09 | 55.52 | 53.89 | 56.71 |
| | DiffAct [45] | 71.15 | 69.72 | 65.45 | 56.76 | 53.86 | 50.63 | 49.12 | *43.32* | *38.86* | 60.62 |
| TSC | MiniRocket [13] | 61.28 | 60.41 | 57.87 | 70.34 | 63.32 | 59.25 | 62.00 | 61.75 | 58.38 | 62.39 |
| | TimesNet [64] | 67.47 | 65.39 | 63.45 | 72.07 | 70.19 | 66.76 | 59.50 | 57.72 | 55.73 | *50.99* |
| | PatchTST [51] | *51.79* | *55.38* | *52.67* | *52.00* | *45.46* | *45.69* | 58.40 | 56.16 | 53.05 | 58.45 |
| NLL | SIGUA [28] | 67.37 | 65.24 | 63.47 | 68.94 | 68.47 | 67.60 | 54.28 | 53.07 | 51.32 | 53.19 |
| | UNICON [36] | 61.15 | 60.45 | 57.35 | 62.26 | 61.63 | 58.34 | 62.26 | 61.63 | 58.34 | 60.53 |
| | Sel-CL [43] | 63.86 | 62.45 | 61.75 | 73.00 | 72.28 | 72.81 | 63.48 | 63.45 | 61.72 | 60.50 |
| TSC & NLL | SREA [9] | 70.10 | 69.65 | 69.40 | 68.64 | 66.02 | 65.67 | *48.81* | 48.80 | 45.72 | 55.21 |
| | Scale-T [47] | 70.40 | 68.06 | 66.51 | 77.77 | 76.71 | **75.97** | 63.21 | 63.40 | 60.77 | 67.64 |
| | *Con4m* | 71.28 | **71.27** | **70.04** | **80.29** | **78.59** | 75.52 | **68.02** | **66.31** | **64.31** | **72.00** |

**Results of different methods.** For fNIRS, TAS models achieve competitive performance compared to *Con4m*, demonstrating the advantage in modeling contextual data dependency among segments. For HHAR, Sleep and SEEG data with more ambiguous boundaries, the performance of TAS models deteriorates significantly, and TSC and NLL models slightly outperform TAS models. Benefiting from multi-scale modeling, Scale-T exhibits significantly better performance on the Sleep and SEEG data compared to SREA. Nevertheless, *Con4m* that fully consider contextual information demonstrate a notable performance improvement (HHAR-0%: 3.24%; Sleep-0%: 7.15%; SEEG: 6.45%) in more complex and ambiguous data.

**Results of different $r\%$.** NLL methods demonstrate close performance degradation as $r\%$ increases from 0% to 20% compared with *Con4m*. However, with a higher ratio from 20% to 40%, SIGUA, UNICON, Sel-CL, SREA, and Scale-T show averaged 3.01%, 5.23%, 1.92%, 3.34%, and 3.22% decrease across fNIRS and Sleep data, while *Con4m* shows 2.37% degradation. For TSC models, non-deep learning-based MiniRocket shows a more robust performance compared to other TSC models. The performance of PatchTST on fNIRS data exhibits significant instability, possibly due to its tendency to overfit inconsistent labels too quickly. DiffAct in TAS models shows the most sensitive performance to boundary perturbations from 0% to 40% across three public data (13.23% decrease). The stable performance of *Con4m* indicates that our proposed training framework can effectively harmonize inconsistent labels.

**Results of symmetric disturbance.** We also corrupt the labels with symmetric disturbance based on segmented labels $y$ rather than raw *MVD* labels, which is commonly employed in the NLL works [62, 43, 32] of the image classification domain. As shown in Figure 3(a), compared to our novel boundary disturbance, *Con4m* exhibits stronger robustness to symmetric disturbance. Even with the 20% disturbance ratio, *Con4m* treats it as a form of data augmentation, resulting in improved performance. This indicates that overcoming more challenging boundary disturbance aligns better with the nature of time series data.

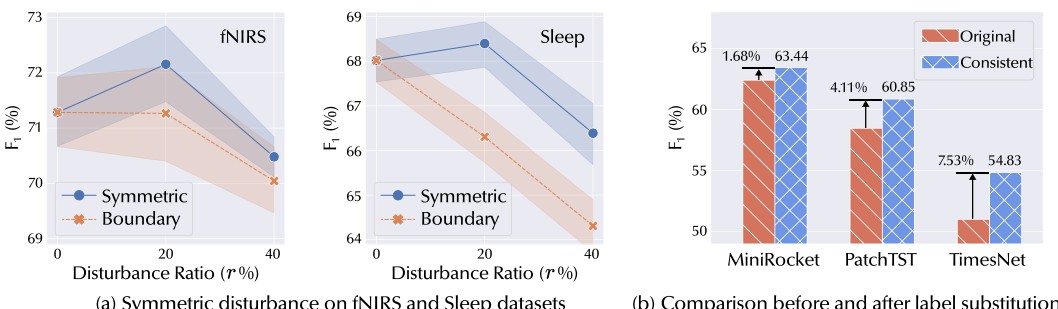

(a) Symmetric disturbance on fNIRS and Sleep datasets    (b) Comparison before and after label substitution

Figure 3: Comparison results of symmetric disturbance and label substitution experiments.

## 4.3 Label Substitution Experiment

Since ambiguous boundaries are inherent to SEEG data and the majority voting procedure is costly, we limit this procedure to only one high-quality testing group in the label disturbance experiment. Besides, on the SEEG data, *Con4m* modifies approximately 10% of the training labels, which is a significant proportion. Therefore, it is necessary to further evaluate the effectiveness of our label harmonization process on SEEG data. Specifically, we train the TSC baselines based on the harmonized labels generated by *Con4m* and observe to what extent the performance of TSC models is improved. As shown in Figure 3(b), PatchTST and TimesNet, employing deep learning architectures, are more susceptible to label inconsistency, so they obtain more significant performance improvement (4.11% and 7.53%). Unlike modified PatchTST that considers the contextual data information across consecutive segments, TimesNet only focuses on the independent segments, thus having a more dramatic improvement. In contrast, MiniRocket achieves only a 1.68% increase, indicating that MiniRocket is more robust with a non-deep learning-based simple random feature mapping.

## 4.4 Ablation Experiment

We introduce two types of model variations. **(1) Preserve only one module.** We preserve only the Con-Transformer (Con-T), Coherent Prediction (Coh-P), or Curriculum Learning (Cur-L) module separately. **(2) Remove only one component.** In addition to removing the above three modules, we also remove the function fitting component (-Fit) and $\eta$ ($E_\eta = 0$) to verify the necessity of prediction behavior constraint and progressively updating labels.

Table 3: Comparison with model ablations in the $F_1$ score (%) in inconsistent scenarios. The **best results** are in bold and we underline the second best results. The *worst results* are denoted in italics.

| Model | | Preserve one | | | | | | Remove one | | | | | | | | | | | Con4m | |
|---|---|---|---|---|---|---|---|---|---|---|---|---|---|---|---|---|---|---|---|---|---|
| | | + Con-T | | + Coh-P | | + Cur-L | | - Con-T | | - Coh-P | | - Cur-L | | - Fit | | - $\eta$ | | | |
| Dataset | $r\%$ | Acc. | $F_1$ | Acc. | $F_1$ | Acc. | $F_1$ | Acc. | $F_1$ | Acc. | $F_1$ | Acc. | $F_1$ | Acc. | $F_1$ | Acc. | $F_1$ | Acc. | $F_1$ |
| Sleep | 20 | 65.97 | 65.05 | 65.76 | 65.10 | 65.31 | 64.76 | 65.73 | 65.53 | 65.84 | 65.07 | 65.85 | 65.43 | 66.06 | 65.28 | *62.02* | *59.97* | **66.61** | **66.31** |
| | 40 | 63.94 | 62.67 | 64.42 | 62.76 | 63.69 | 62.23 | 64.44 | 63.05 | 64.23 | 63.03 | 64.89 | 63.07 | 64.69 | 63.22 | *61.93* | *57.98* | **65.34** | **64.31** |
| SEEG | - | 71.68 | 67.85 | 71.69 | 69.04 | 71.32 | 67.22 | 73.85 | 70.59 | 72.41 | 68.26 | 74.17 | 71.18 | 73.47 | 70.63 | *70.70* | *66.04* | **74.60** | **72.00** |

As shown in Table 3, when keeping one module, +Coh-P achieves the best performance with an averaged 2.78% decrease in $F_1$ score, indicating that introducing the contextual label information are most effective for *MVD*. The utility of each module varies across datasets. For example, for Sleep data, the Con-T contributes more to performance improvement compared to the Cur-L module, while the opposite phenomenon is observed for SEEG data. As for removing one component, even when we only remove the Tanh function fitting, the $F_1$ score significantly decreases 1.72% on average. On

the Sleep-20% and SEEG data, the drop caused by -Fit is more significant than that caused by some other modules. Moreover, the model variation -$\eta$ achieves the worst results ($9.23\%$ decrease in $F_1$). The results imply that during early training stages, the model tends to learn the consistent parts of the raw labels. Premature use of unreliable predicted labels as subsequent training supervision signals leads to model poisoning and error accumulation.

### 4.5   Case Study

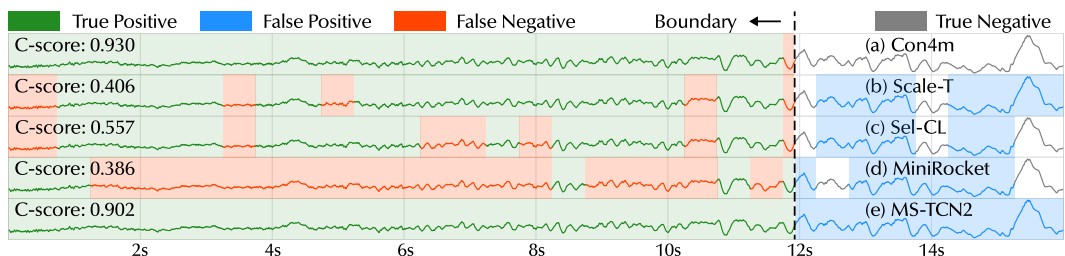

Figure 4: Case study for a continuous time interval in SEEG testing set. The C-score, introduced by the ClaSP model [19], assessing the ability of models to recognize segmentation boundaries by measuring the trade-off between precision (correctly identified change points) and recall (finding all true change points).

We present a case study to provide a specific example that illustrates how *Con4m* works for *MVD* in Figure 4. We show comparative visualization results for the predictions in a continuous time interval in the SEEG testing set. In SEEG data, we assign the label of normal segments as $0$ and that of seizures as $1$. As the figure shows, *Con4m* demonstrates a more coherent narrative by constraining the prediction behavior and aligning with the contextual data information. In contrast, Scale-T, Sel-CL and MiniRocket exhibit noticeably interrupted and inconsistent predictions. MS-TCN2 fails to identify normal segments. More impressively, *Con4m* accurately identifies the consistent boundary within the time interval spanning across two classes. We also utilize the C-score proposed by the ClaSP model [19] to assess the segmentation capability of models. The C-score reflects the ability of models to recognize segmentation boundaries by measuring the trade-off between precision (correctly identified change points) and recall (finding all true change points), ensuring the model captures meaningful transitions without over-segmenting or missing important splits. We compute the scores for each model across three sets of experiments on SEEG data and take the average. Notably, *Con4m* outperforms the other models significantly, while MS-TCN2, specifically designed for segmentation tasks, also achieves impressive scores. This verifies that the label consistency framework can harmonize the boundaries more effectively. Refer to Appendix H for more cases.

## 5   Conclusion and Discussion

In this work, we focus on the raw time series *MVD* for segmented time series classification (TSC) tasks, demonstrating unique challenges that are overlooked by existing mainstream TSC models. We first formally demonstrate that valuable contextual information enhances the discriminative power of classification instances. Based on the insights, we introduce contextual prior knowledge of data locality and label coherence to guide the model to focus on contextual information more conducive to discriminating consecutive segments in segmented TSC tasks. Leveraging effective contextual information, a label consistency learning framework *Con4m* is proposed to progressively harmonize inconsistent labels during training. Extensive experiments validate the superior performance achieved by *Con4m* and highlight the effectiveness of the proposed consistent label training framework. Our work still has some limitations. We have solely focused on analyzing and designing end-to-end supervised models. Further exploration of large-scale models would be challenging yet intriguing. *Con4m* is a combination of segmentation and classification, both of which are fully supervised. Exploring its application in unsupervised segmentation tasks is worthwhile. When faced with more diverse label behaviors, the function fitting module needs to engage in more selection and design of basis functions. Nevertheless, our work brings new insights to the TSC domain, re-emphasizing the importance of the inherent temporal dependence of time series.

## Acknowledgments and Disclosure of Funding

This work was partially supported by National Natural Science Foundation of China (No. 62322606, No. 62441605).

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

# A Details of Related Works

**Time series classification (TSC).** TSC has become a popular field in various applications with the exponential growth of available time series data in recent years. In response, researchers have proposed numerous algorithms [34]. High accuracy in TSC is achieved by classical algorithms such as Rocket and its variants [12, 13], which use random convolution kernels with relatively low computational cost, as well as ensemble methods like HIVE-COTE [44], which assign weights to individual classifiers.

Moreover, the flourishing non-linear modeling capacity of deep models has led to an increasing prevalence of TSC algorithms based on deep learning. Various techniques are utilized in TSC: RNN-based methods [54, 14] capture temporal changes through state transitions; MLP-based methods [22, 65] encode temporal dependencies into parameters of the MLP layer; and the latest method TimesNet [64] converts one-dimensional time series into a two-dimensional space, achieving state-of-the-art performance on five mainstream tasks. Furthermore, Transformer-based models [67, 10] with attention mechanism have been widely used.

The foundation of our work lies in these researches, including the selection of the backbone and experimental setup. However, mainstream TSC models [50, 20] are often designed for publicly available datasets [3, 11] based on the *i.i.d.* samples, disregarding the inherent contextual dependencies between classified segments in *MVD*. Although some time series models [55, 51] use patch-by-patch technique to include contextual information, they are partially context-aware since they only model the data dependencies within each time segment, ignoring the dependencies between consecutive segments.

**Noisy label learning (NLL).** NLL is an important and challenging research topic in machine learning, as real-world data often rely on manual annotations prone to errors. Early works focus on statistical learning [1, 41, 5]. Researches including Sukhbaatar et al. [57] launch the era of noise-labeled representation learning.

The label noise transition matrix, which represents the transition probability from clean labels to noisy labels [29], is an essential tool. Common techniques for loss correction include forward and backward correction [53], while masking invalid class transitions with prior knowledge is also an important method [26]. Adding an explicit or implicit regularization term in objective functions can reduce the model's sensitivity to noise, whereas re-weighting mislabeled data can reduce its impact on the objective [2, 69, 46]. Other methods involve training on small-loss instances and utilizing memorization effects. MentorNet [35] pretrains a secondary network to choose clean instances for primary network training. Co-teaching [27] and Co-teaching+ [70], as sample selection methods, introduce two neural networks with differing learning capabilities to train simultaneously, which filter noise labels mutually. The utilization of contrastive learning has emerged as a promising approach for enhancing the robustness in the context of classification tasks of label correction methods [43, 72, 32].

These works primarily focus on handling noisy labels. And ensuring overall label consistency by modifying certain labels is crucial for *MVD*. To the best of our knowledge, Scale-teaching [47] and SREA [9] are the only NLL works specifically designed for time series. Scale-teaching designs a fine-to-coarse cross-scale fusion mechanism for learning discriminative patterns by utilizing time series at different scales to train multiple DNNs simultaneously. SREA trains a classifier and an autoencoder with a shared embedding representation, progressively self-relabeling mislabeled data samples in a self-supervised manner. However, they still face the issue of overlooking contextual dependencies across consecutive time segments.

**Curriculum learning (CL).** Bengio et al. [6] propose CL, which imitates human learning by starting with simple samples and progressing to complicated ones. Based on this notion, CL can denoise noisy data since learners are encouraged to train on easier data and spend less time on noisy samples [24, 60]. Current mainstream approaches include Self-paced Learning [40], where students schedule their learning, Transfer Teacher [63], based on a predefined training scheduler; and RL Teacher [25, 48], which incorporates student feedback into the framework. The utilization of CL proves to be particularly advantageous in situations involving changes in the training labels. Hence, this technique is utilized to enhance the harmonization process of boundary labels from *MVD* in a more stable manner.

**Temporal action segmentation (TAS).** TAS is a critical task in video understanding and analysis. It involves segmenting an untrimmed video sequence into meaningful temporal segments and assigning a

predefined action label to each segment [17]. The majority works on TAS typically take visual feature vectors, either hand-crafted (IDT) [59] or extracted from an off-the-shelf CNN backbone (I3D) [8], as input for each frame. TAS uses sequential modeling to incorporate temporal dependencies and sequential context for improved TAS accuracy.

While TAS shares some similarities with segmented TSC, there still exist some problems. Time series data often have high sampling rates, however, it is impossible to input excessively long sequences into TAS models designed for video data. Moreover, unlike the I3D features [8] used as input in these works, time series lack a unified pretrained model for feature extraction. Besides, the changes between adjacent video frames are smooth and continuous, whereas time series are more variable and exhibit more ambiguous boundaries. Also, TAS works focus on modeling the dependency of instances from a data perspective without explicitly leveraging contextual label information.

## B    Implementation Details of Prediction Behavior Constraint

To fit the hyperbolic tangent function (Tanh), we use the mean squared error (MSE) loss function. In practice, we use the Adam optimizer with a learning rate of $0.1$ to optimize the trainable parameters. The maximum number of iterations is set to $100$, and the tolerance value for stopping the fitting process based on loss change is set to $1e-6$. Sequences belonging to one minibatch are parallelized to fit their respective Tanh functions. To adapt to the value range of the standard Tanh function, we rescale the sequential predictions to $[-1, 1]$ before fitting.

However, it can be difficult to achieve a good fit when fitting with the Tanh function. Specifically, random initialization may fail to fit the sequential values properly when a long time series undergoes a state transition near the boundary. For example, as Figure 5(a) shows, we fit a sequence in which only the last value is $1$. We set all default initial parameters as $1$ and fit it. It can be observed that the fitting function cannot properly fit the trend and will mislabel the last point.

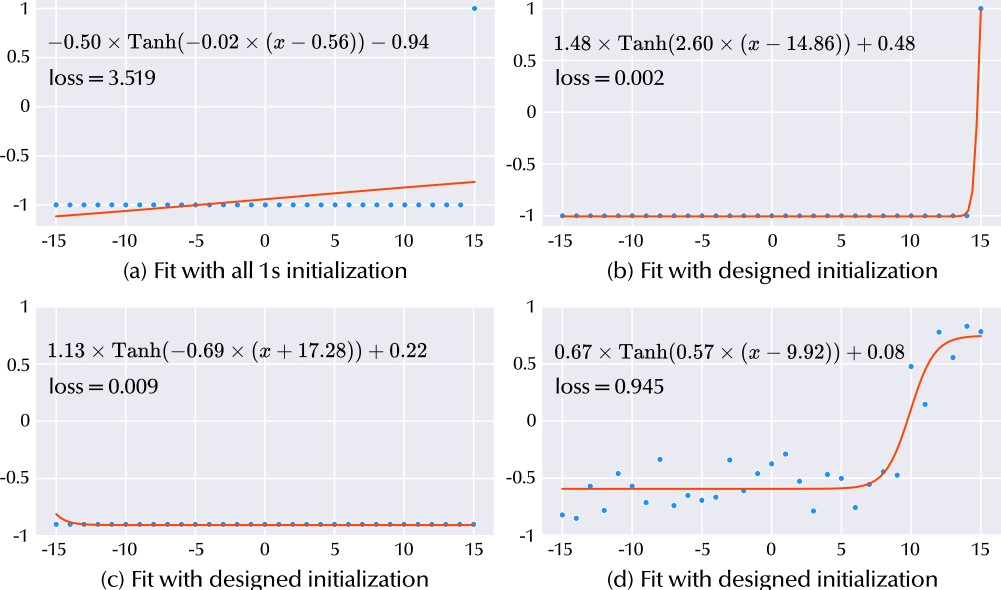

Figure 5:  Cases for Tanh fitting.

Appropriate parameter initialization is needed to avoid excessive bias. After careful observation, we find that parameter $k$ controls the slope at the transition part of Tanh, and parameter $b$ controls the abscissa at the transition point. In the process, all fitting values are assigned with uniform abscissa values. Therefore, we calculate the maximum difference between adjacent values and the corresponding position in the entire sequence. And these two values are assigned to parameters $k$ and $b$, respectively. This allows us to obtain suitable initial parameters and avoid getting trapped in local optima or saddle points during function fitting. Formally, given the $L$-length input sequence $\tilde{p}$, we initialize parameters $k$ and $b$ as follows:

$$di = [\tilde{p}_{i+1} - \tilde{p}_i]_{i \in \{1, \ldots, L-1\}},$$

$$k, b = \max\left(\mathrm{Abs}(di)\right), \arg\max\left(\mathrm{Abs}(di)\right),$$
$$k = k \times \mathrm{Sign}(di[b]),$$
$$b = -\left(b - \lfloor L/2 \rfloor + 0.5\right),$$

where $\mathrm{Abs}(\cdot)$ and $\mathrm{Sign}(\cdot)$ denote the absolute value function and sign function respectively. $di$ is the difference vector. After proper initialization, as Figure 5(b) shows, we can obtain more accurate fitting results to reduce the probability of mislabeling. We also show some other cases (Figure 5(c)(d)) for the fitting results to verify the effectiveness of the fitting process we propose.

## C  Hyperparameter Analysis

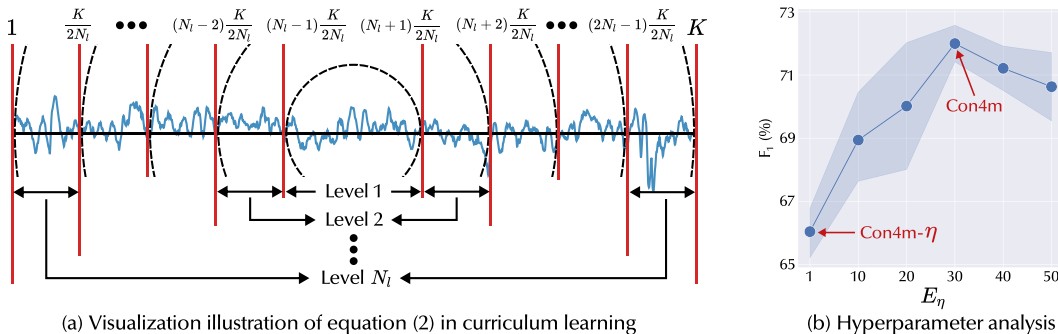

(a) Visualization illustration of equation (2) in curriculum learning  (b) Hyperparameter analysis

Figure 6: Visualization of data division in curriculum learning and hyperparameter analysis of $E_\eta$.

The dynamic weighting factor $\eta$ is introduced to progressively update the labels, preventing the model from overly relying on its own predicted labels too early. To validate the utility of $\eta$ and determine an appropriate linear growth epoch $E_\eta$, we conduct the hyperparameter search experiment on SEEG data. As shown in Figure 6(b), with smaller $E_\eta$ (corresponding to a higher growth rate), there is a significant improvement in model performance. This aligns with our motivation that during the early stage of model training, the primary objective is to better fit the original labels. At this stage, the model's own predictions are unreliable. If the predicted results are used as training labels too early in subsequent epochs, the model would be adversely affected by its own unreliability. On the other hand, excessively large $E_\eta$ leads to a slower rate of label updates, making it more challenging for the model to timely harmonize inconsistent labels. Nonetheless, considering the impact of variance, the model exhibits robustness to slightly larger $E_\eta$. In this work, we uniformly use $E_\eta = 30$ as the default value.

## D  Details of Datasets

**fNIRS.** All signals are sampled at a frequency of $5.2$Hz. At each time step, they record $8$ real-valued measurements, with each measurement corresponding to $2$ concentration changes (oxyhemoglobin and deoxyhemoglobin), $2$ types of optical data (intensity and phase), and $2$ spatial positions on the forehead. Each measurement unit is a micromolar concentration change per liter of tissue (for oxy-/deoxyhemoglobin). They label each part of the active experiment with one of four possible levels of $n$-back working memory intensity (0-back, 1-back, 2-back, or 3-back). More specifically, in an $n$-back task, the subject receives $40$ numbers in sequence. If a number matches the number $n$ steps back, the subject is required to respond accordingly. There are $16$ rounds of tasks, with a 20-second break between each task. Following Huang et al. [31], we only apply classification tasks for 0-back and 2-back tasks in our work. Therefore, we only extract sequences for 0-back and 2-back tasks and concatenate them in chronological order.

**HHAR.** The HHAR (Heterogeneity Human Activity Recognition) dataset [7] captures sensor data from multiple smart devices to explore the impact of device heterogeneity on human activity recognition. It involves measurements from accelerometers and gyroscopes, which are two common motion sensors found in smartphones and smartwatches. The dataset includes six types of human activities: 'Biking', 'Sitting', 'Standing', 'Walking', 'Stairs Up', and 'Stairs Down'. For each activity, both 3-axis accelerometer and 3-axis gyroscope readings are recorded, resulting in six key features

per time step. The data was collected from five different device types, consisting of four smartwatches and eight smartphones, across nine participants. Each device samples data at its highest possible frequency, generally around 50Hz. For data preprocessing, the accelerometer and gyroscope signals are aligned based on their timestamps, normalized within each channel, and divided by the devices, following the procedure outlined in Gagnon-Audet et al. [21]. To ensure balanced samples, we combined data from the Galaxy S3 Mini, LG watch, and Gear watch into a single group.

**Sleep.** The Sleep-EDF database records PolySomnoGraphic sleep data from 197 subjects, including EEG, EOG, chin EMG, and event markers. Some data also includes respiration and temperature-related signals. The database contains two studies: the Sleep Cassette study and the Sleep Telemetry study. The former records approximately 40 hours of sleep from two consecutive nights, while the latter records around 18 hours of sleep. Well-trained technicians manually score the corresponding sleep graphs according to the Rechtschaffen and Kales manual. The data is labeled in intervals of 30 seconds, with each interval being marked as one of the eight possible stages: W, R, 1, 2, 3, 4, M, or ?. In our work, we utilize only the data from the Sleep Cassette study, and retain only the signals from the EEG Fpz-Cz channel and EOG horizontal channel. The EEG and EOG signals were sampled at a frequency of 100Hz. Following Kemp et al. [37], we remove the labels for stages ? and M from the data, and merge stages 3 and 4, resulting in a 5-classification task.

**SEEG.** The private SEEG data records brain signals indicative of suspected pathological tissue within the brains of seizure patients. They are anonymously collected from a top hospital we cooperate with. For a patient suffering from epilepsy, 4 to 11 invasive electrodes with 52 to 153 channels are used for recording signals. In total, we have collected 847 hours of SEEG signals with a high frequency (1,000Hz or 2,000Hz) and a total capacity of 1.2TB. Professional neurosurgeons help us label the seizure segments for each channel. Before sampling for the database, we remove the bad channels marked by neurosurgeons. Then we uniformly downsample the data to 250Hz and use a low-pass filter to process the data with a cutoff frequency of 30Hz. Finally, we normalize and sample the intervals for each channel respectively.

## E    Implementation Details of Baselines

- **SREA** [9]: This time series classification model with noisy labels jointly trains a classifier and an autoencoder with shared embedding representations. It gradually corrects the mislabelled data samples during training in a self-supervised fashion. We use the default model architecture from the source code provided by the author (https://github.com/Castel44/SREA).

- **Scale-teaching** [47]: This work designs a fine-to-coarse cross-scale fusion mechanism for learning discriminative patterns by utilizing time series at different scales to train multiple DNNs simultaneously. It uses well-learned multi-scale time series embeddings for noise label correction at sample feature levels. We modify the code to match our datasets based on the code provided by the author (https://github.com/qianlima-lab/Scale-teaching).

- **SIGUA** [28]: This model adopts gradient descent on good data as usual, and learning-rate-reduced gradient ascent on bad data, thereby trying to reduce the effect of noisy labels. We modify the network for time series data based on the open source code provided by SREA, using the code from the author (https://github.com/bhanML/SIGUA).

- **UNICON** [36]: UNICON introduces a Jensen-Shannon divergence-based uniform selection mechanism and uses contrastive learning to further combat the memorization of noisy labels. We modify the model for time series data according to the code provided by the author (https://github.com/nazmul-karim170/UNICON-Noisy-Label)

- **Sel-CL** [43]: Selective-Supervised Contrastive Learning (Sel-CL) is a latest baseline model in the field of computer vision. It selects confident pairs out of noisy ones for supervised contrastive learning (Sup-CL) without knowing noise rates. We modify the code for time series data, based on the source code provided by the author (https://github.com/ShikunLi/Sel-CL)

- **MiniRocket** [13]: Rocket [12] achieves state-of-the-art accuracy for time series classification by transforming input time series using random convolutional kernels, and using the transformed features to train a linear classifier. MiniRocket is a variant of Rocket that improves processing time, while offering essentially the same accuracy. We use the code interface from the sktime package (https://github.com/sktime/sktime).

- **TimesNet** [64]: This model focuses on temporal variation modeling. With TimesBlock, it can discover the multi-periodicity adaptively and extract the complex temporal variations from transformed 2D tensors by a parameter-efficient inception block. We use the code from the TSlib package (https://github.com/thuml/Time-Series-Library).

- **PatchTST** [51]: This is a self-supervised representation learning framework for multivariate time series by segmenting time series into subseries level patches, which are served as input tokens to Transformer with channel-independence. We modify the code to achieve classification for each patch, based on the source code from the Time Series Library (TSlib) package (https://github.com/thuml/Time-Series-Library).

- **MS-TCN2** [42]: This work proposes two multi-stage architectures for the temporal action segmentation task. While the first stage generates an initial prediction, this prediction is iteratively refined by the higher stages. Instead of the commonly used temporal pooling, they use dilated convolutions to increase the temporal receptive field. We modify the code for time series data, based on the source code provided by the author (https://github.com/sj-li/MS-TCN2).

- **ASFormer** [68]: ASFormer is a Transformer-based model for action segmentation tasks. It explicitly brings in the inductive priors of local connectivity and applies a pre-defined hierarchical representation pattern to handle long input sequences. The decoder is also carefully designed to refine the initial prediction from the encoder. We modify the code for time series data, based on the source code provided by the author (https://github.com/ChinaYi/ASFormer).

- **DiffAct** [45]: In this work, action predictions are iteratively generated from random noise with input video features as conditions. It also devises a unified masking strategy for the conditioning inputs to enhance the modeling of striking characteristics of human actions. We modify the code for time series data, based on the source code provided by the author (https://github.com/Finspire13/DiffAct).

# F    Implementation Details of *Con4m*

The non-linear encoder $g_{enc}$ used in *Con4m* is composed of three 1-D convolution layers. The number of kernels vary across different data and you can find corresponding parameters in the default config file of our source code. We construct the Con-Transformer based on the public codes implemented by HuggingFace[1]. We set $d$=128 and the dimension of intermediate representations in FFN module as 256 for all experiments. The number of heads and dropout rate are set as 8 and 0.1 respectively. Since we observe that one-layer Con-Attention can fit the data well, we do not stack more layers to avoid overfitting. Note that *Con4m* consists of two Con-Transformers and we use two Con-Attention layers. The model is optimized using Adam optimizer [38] with a learning rate of $1e$-3 and weight decay of $1e$-4, and the batch size is set as 64. We build our model using PyTorch 2.0.0 [52] with CUDA 11.8. And the model is trained on a workstation (Ubuntu system 20.04.5) with 2 CPUs (AMD EPYC 7H12 64-Core Processor) and 8 GPUs (NVIDIA GeForce RTX 3090).

# G    Full Results

The full results of the label disturbance experiment are listed in Table 4, 5, 6 and 7. For fNIRS, we first divide the data into 4 groups by subjects and follow the 2 training-1 validation-1 testing (2-1-1) setting to conduct cross-validation experiments. Therefore, there are $C_4^2 \times C_2^1 = 12$ experiments in total. Similarly, we divide the HHAR and Sleep data into 3 groups and follow the 1-1-1 experimental setting. Therefore, we carry out $C_3^1 \times C_2^1 = 6$ experiments. For SEEG data, we follow the same setting as fNIRS. Notice that for SEEG data, inconsistent labels already exist in the raw data. We obtain a high-quality testing group by using a majority voting procedure to determine the boundaries. Then we leave the testing group aside and only change the validation group to report the mean value of $C_3^2 = 3$ experiments. All the experimental results are listed in lexicographical order according to the group name composition. We also report the mean value and standard derivation of all experiments. Specifically, we use the STDEVA to estimate standard deviation based on a sample of data.

---

[1]https://github.com/huggingface/transformers/blob/v4.25.1/src/transformers/models/bert/modeling_bert.py

| $r\%$ | Exp | TAS | | | TSC | | | NLL | | | TSC&NLL | | |
|---|---|---|---|---|---|---|---|---|---|---|---|---|---|
| | | MS-TCN2 | AS Former | DiffAct | Mini Rocket | Times Net | Patch TST | SI-GUA | UNI-CON | Sel-CL | SREA | Scale-T | Con4m |
| 0 | 1 | 68.61 | 69.76 | **71.11** | 61.37 | 60.73 | *51.07* | 64.75 | 63.85 | 63.95 | 69.13 | 66.07 | 68.55 |
| | 2 | 70.50 | 71.85 | 68.62 | 62.45 | 68.25 | *48.21* | 67.55 | 57.17 | 65.49 | 70.29 | **74.26** | 71.64 |
| | 3 | 70.21 | 70.08 | 71.42 | 60.96 | 66.38 | *54.23* | 65.56 | 61.71 | 61.44 | 67.14 | 68.99 | 70.51 |
| | 4 | 71.89 | **73.38** | 72.25 | 62.24 | 69.73 | *55.07* | 68.83 | 60.74 | 63.23 | 70.23 | 71.64 | 72.65 |
| | 5 | 71.87 | 72.91 | 72.80 | 61.35 | 66.46 | 57.11 | 67.96 | *54.87* | 63.21 | 70.13 | **75.85** | 68.55 |
| | 6 | 72.53 | **73.12** | 71.29 | 61.79 | 69.58 | *57.22* | 70.12 | 58.96 | 64.66 | 69.66 | 70.50 | 72.99 |
| | 7 | 69.08 | 68.08 | 70.77 | 60.11 | 62.64 | *50.46* | 64.03 | 62.54 | 60.13 | 70.55 | 62.42 | 70.63 |
| | 8 | 72.29 | 72.44 | 72.13 | 62.90 | 69.57 | *49.75* | 69.15 | 63.05 | 65.31 | 70.90 | 66.81 | **73.36** |
| | 9 | 72.52 | 72.22 | 69.47 | 58.78 | 67.23 | *48.86* | 68.41 | 66.77 | 63.45 | 70.29 | 71.88 | 72.60 |
| | 10 | **74.48** | 70.20 | 73.02 | 60.75 | 71.17 | *55.63* | 68.24 | 59.40 | 65.47 | 71.59 | 70.60 | 69.38 |
| | 11 | 70.71 | 71.23 | 70.09 | 60.71 | 66.64 | *48.51* | 65.95 | 57.84 | 65.24 | 69.17 | **76.77** | 69.42 |
| | 12 | 73.08 | 75.02 | 70.81 | 61.93 | 71.30 | *45.40* | 67.89 | 66.88 | 64.70 | 72.17 | 68.95 | **75.14** |
| | Avg | 71.48 | **71.69** | 71.15 | 61.28 | 67.47 | *51.79* | 67.37 | 61.15 | 63.86 | 70.10 | 70.40 | 71.28 |
| | Std | 1.70 | 1.91 | 1.32 | **1.13** | 3.23 | 3.91 | 1.87 | 3.72 | 1.69 | 1.28 | *4.14* | 2.11 |
| 20 | 1 | 67.10 | 70.19 | 72.07 | 59.22 | 62.74 | *49.81* | 61.42 | 64.38 | 60.91 | 68.32 | 65.55 | 69.48 |
| | 2 | 71.26 | 70.42 | 69.61 | 60.10 | 67.31 | *58.02* | 63.38 | 63.30 | 64.00 | 70.40 | 68.36 | **72.94** |
| | 3 | 72.18 | 66.46 | 71.39 | 59.52 | 60.19 | 54.99 | 64.27 | *51.67* | 57.25 | 70.16 | 64.44 | 72.06 |
| | 4 | 71.19 | 71.74 | 71.53 | 63.15 | 66.04 | 62.28 | 67.91 | 62.23 | *61.45* | 69.78 | 66.18 | **73.56** |
| | 5 | 71.85 | **72.38** | 69.18 | 62.04 | 67.66 | 55.29 | 66.07 | 56.26 | 64.22 | 68.45 | 66.59 | 71.41 |
| | 6 | 72.03 | 69.99 | 69.93 | 61.58 | 68.32 | *57.22* | 67.09 | 62.59 | 65.45 | 70.34 | 68.71 | **72.08** |
| | 7 | 69.94 | 66.99 | 68.32 | 59.15 | 59.02 | 53.13 | 60.97 | *49.65* | 59.18 | 67.69 | **71.55** | 62.68 |
| | 8 | 69.19 | 72.09 | 64.24 | 60.33 | 66.27 | 56.57 | 63.72 | 66.42 | 63.28 | 70.85 | **75.03** | 71.59 |
| | 9 | **74.44** | 72.21 | 69.70 | 59.41 | 66.73 | 52.13 | 67.11 | 62.28 | 61.54 | 68.15 | 66.94 | 71.84 |
| | 10 | 72.15 | **72.79** | 69.91 | 60.77 | 69.07 | 56.96 | 68.27 | 59.87 | 65.18 | 71.29 | 64.83 | 71.72 |
| | 11 | 66.50 | 73.26 | 68.58 | 58.87 | 65.17 | *49.62* | 64.83 | 63.95 | 63.32 | 69.48 | 68.88 | 72.08 |
| | 12 | **74.10** | 70.41 | 72.16 | 60.79 | 66.22 | 58.49 | 67.87 | 62.81 | 63.62 | 70.93 | 69.65 | 73.76 |
| | Avg | 70.99 | 70.75 | 69.72 | 60.41 | 65.39 | *55.38* | 65.24 | 60.45 | 62.45 | 69.65 | 68.06 | **71.27** |
| | Std | 2.45 | 2.17 | 2.16 | 1.32 | 3.15 | 3.71 | 2.53 | *5.22* | 2.46 | **1.22** | 3.03 | 2.92 |
| 40 | 1 | 68.74 | 65.53 | 65.47 | 57.21 | 62.93 | *51.39* | 60.63 | 52.63 | 61.98 | 69.37 | 62.03 | 65.90 |
| | 2 | 67.96 | 70.88 | 64.39 | 58.85 | 62.10 | 50.27 | 59.84 | *45.74* | 62.50 | 69.43 | 68.10 | **71.91** |
| | 3 | 68.25 | 68.40 | 68.79 | 58.30 | 60.06 | *44.09* | 65.00 | 54.70 | 59.58 | 69.12 | 64.79 | **71.05** |
| | 4 | 70.76 | **71.54** | 66.20 | 59.23 | 68.56 | 58.48 | 67.18 | 63.46 | 64.25 | 68.84 | 66.90 | 70.68 |
| | 5 | 72.62 | 66.59 | 64.44 | 57.05 | 59.96 | 54.44 | 64.60 | 60.00 | 58.31 | 68.49 | 67.67 | **71.55** |
| | 6 | 71.36 | 72.64 | 61.00 | 58.43 | 66.76 | 53.18 | 64.20 | 59.95 | 61.72 | 69.85 | 68.12 | **72.75** |
| | 7 | 67.31 | 67.94 | 68.49 | 56.25 | 60.06 | *49.93* | 61.58 | 52.33 | 60.33 | **69.19** | 59.47 | 66.69 |
| | 8 | 66.67 | 66.67 | 65.84 | 58.26 | 68.32 | *53.20* | 67.27 | 60.76 | 63.91 | **70.49** | 68.52 | 68.50 |
| | 9 | 66.59 | 67.74 | 66.30 | *56.95* | 63.86 | 61.90 | 61.80 | 65.20 | 62.82 | 68.42 | 67.82 | **69.88** |
| | 10 | 69.56 | 66.67 | 65.55 | 55.78 | 62.01 | *49.37* | 63.19 | 55.84 | 61.04 | **70.16** | 66.68 | 69.00 |
| | 11 | 71.66 | **72.80** | 62.43 | 58.34 | 62.78 | *57.77* | 62.12 | 57.92 | 61.68 | 68.36 | 68.64 | 70.59 |
| | 12 | 71.32 | **72.74** | 66.53 | 59.81 | 63.96 | *48.00* | 64.21 | 56.73 | 62.93 | 71.05 | 69.38 | 72.02 |
| | Avg | 69.40 | 69.18 | 65.45 | 57.87 | 63.45 | *52.67* | 63.47 | 57.35 | 61.75 | 69.40 | 66.51 | **70.04** |
| | Std | 2.10 | 2.75 | 2.22 | 1.22 | 3.03 | 4.95 | 2.38 | *5.57* | 1.74 | **0.85** | 2.98 | 2.14 |

Table 4: Full results of the label disturbance experiment on **fNIRS** data. The **best results** are in bold and we underline the second best results. The *worst* results are denoted in italics.

## H  Case Study

As shown in Figure 7, we present four cases to compare and demonstrate the differences between our proposed *Con4m* and other baselines. The first two cases involve transitions from a seizure state of label 1 to a normal state of label 0. The third case consists of entirely normal segments, while the fourth case comprises entirely seizure segments. As illustrated in the figure, *Con4m* exhibits more coherent narratives by constraining the predictions to align with the contextual information of the data. Moreover, it demonstrates improved accuracy in identifying the boundaries of transition states. In contrast, other baselines exhibit fragmented and erroneous predictions along the time segments. This verifies that *Con4m* can achieve clearer recognition of boundaries, and it can also make better predictions on the consecutive time segments belonging to the same class.

| r% | Exp | TAS | | | TSC | | | NLL | | | TSC&NLL | | |
|---|---|---|---|---|---|---|---|---|---|---|---|---|---|
| | | MS-TCN2 | AS Former | DiffAct | Mini Rocket | Times Net | Patch TST | SI-GUA | UNI-CON | Sel-CL | SREA | Scale-T | Con4m |
| 0 | 1 | 41.39 | 34.82 | 41.92 | **52.89** | 45.80 | *30.88* | 46.79 | 52.71 | 47.69 | 47.40 | 45.80 | 52.37 |
| | 2 | 77.19 | 55.28 | 53.24 | 75.76 | 89.33 | *52.88* | 80.21 | **98.20** | 93.87 | 76.48 | 95.08 | 95.04 |
| | 3 | 43.71 | *42.61* | 45.85 | **54.11** | 44.75 | 42.92 | 46.89 | 47.60 | 46.38 | 48.14 | 49.43 | 48.45 |
| | 4 | 81.92 | 82.37 | 65.32 | 75.00 | 92.82 | *55.04* | 77.44 | 96.50 | 81.12 | 81.04 | 93.65 | **98.78** |
| | 5 | 90.67 | 79.96 | *70.51* | 94.37 | 88.03 | 70.61 | 94.05 | 93.02 | 87.18 | 91.70 | **94.63** | 94.09 |
| | 6 | 83.89 | 80.07 | 63.73 | 69.91 | 71.70 | *59.66* | 68.27 | 83.61 | 81.73 | 67.06 | 88.05 | **93.02** |
| | Avg | 69.79 | 62.52 | 56.76 | 70.34 | 72.07 | *52.00* | 68.94 | 78.61 | 73.00 | 68.64 | 77.77 | **80.29** |
| | Std | 21.56 | 21.08 | **11.51** | 15.47 | 22.00 | 13.74 | 19.01 | 22.67 | 20.63 | 18.01 | *23.52* | 23.26 |
| 20 | 1 | 42.39 | 44.86 | 38.64 | 47.97 | 45.96 | *30.52* | 47.59 | 51.25 | 46.59 | 45.54 | 48.25 | **54.82** |
| | 2 | 68.54 | 64.63 | 54.97 | 71.37 | 83.16 | *39.71* | 81.73 | **97.72** | 90.03 | 70.52 | 95.35 | 93.31 |
| | 3 | 46.72 | 47.95 | *38.36* | 49.72 | 48.64 | 43.00 | 51.86 | 50.40 | 47.72 | **53.70** | 51.06 | 45.06 |
| | 4 | 87.95 | 77.25 | 66.51 | 69.95 | 85.91 | *41.34* | 76.36 | 91.20 | 86.85 | 73.44 | 93.33 | **96.37** |
| | 5 | 82.97 | *68.53* | 69.22 | 86.79 | 87.36 | 68.77 | 88.61 | **91.35** | 89.62 | 85.33 | 90.48 | 90.61 |
| | 6 | 71.76 | 62.32 | 55.47 | 54.14 | 70.13 | *49.41* | 64.65 | 79.70 | 72.84 | 67.55 | 81.81 | **91.38** |
| | Avg | 66.72 | 60.92 | 53.86 | 63.32 | 70.19 | *45.46* | 68.47 | 76.94 | 72.28 | 66.02 | 76.71 | **78.59** |
| | Std | 18.63 | **12.38** | 13.20 | 15.26 | 18.77 | 12.95 | 16.55 | 21.05 | 20.45 | 14.29 | 21.48 | *22.49* |
| 40 | 1 | 39.17 | 42.50 | 37.82 | 47.65 | 44.86 | *30.87* | **52.92** | 48.83 | 48.60 | 46.60 | 50.21 | 50.78 |
| | 2 | 59.62 | 59.00 | 51.47 | 66.27 | 73.60 | *42.93* | 72.53 | 92.05 | **94.60** | 67.78 | 94.41 | 91.36 |
| | 3 | 54.82 | 46.54 | 42.00 | 48.52 | 43.99 | *39.89* | 52.23 | 49.13 | 47.87 | **55.59** | 48.69 | 53.33 |
| | 4 | 72.94 | 78.15 | 60.46 | 62.46 | 87.26 | *46.10* | 76.04 | 75.99 | 80.98 | 74.54 | **93.49** | 91.57 |
| | 5 | 75.12 | 75.85 | *57.81* | 77.54 | 79.70 | 62.93 | 88.42 | **93.46** | 82.62 | 85.48 | 87.43 | 82.62 |
| | 6 | 72.08 | 62.57 | 54.23 | 53.03 | 71.13 | *51.42* | 63.43 | 79.79 | 78.15 | 64.05 | 81.56 | **83.48** |
| | Avg | 62.29 | 60.77 | 50.63 | 59.25 | 66.76 | *45.69* | 67.60 | 73.21 | 72.81 | 65.67 | **75.97** | 75.52 |
| | Std | 13.94 | 14.64 | **8.95** | 11.68 | 18.18 | 10.87 | 14.13 | 19.95 | 19.85 | 13.74 | *21.06* | 18.58 |

Table 5: Full results of the label disturbance experiment on **HHAR** data. The **best results** are in bold and we underline the second best results. The *worst* results are denoted in italics.

| r% | Exp | TAS | | | TSC | | | NLL | | | TSC&NLL | | |
|---|---|---|---|---|---|---|---|---|---|---|---|---|---|
| | | MS-TCN2 | AS Former | DiffAct | Mini Rocket | Times Net | Patch TST | SI-GUA | UNI-CON | Sel-CL | SREA | Scale-T | Con4m |
| 0 | 1 | 59.46 | 55.89 | 51.79 | 62.16 | 58.73 | 58.42 | 54.79 | 62.41 | 63.49 | *48.95* | 63.05 | **68.80** |
| | 2 | 59.47 | 58.37 | *39.55* | 61.14 | 59.72 | 58.60 | 52.69 | 62.49 | 62.87 | 46.93 | 62.70 | **67.63** |
| | 3 | 60.72 | 59.60 | 52.65 | 62.74 | 60.76 | 59.44 | 56.19 | 62.62 | 65.47 | *49.38* | 64.91 | **69.29** |
| | 4 | 61.41 | 58.65 | *47.42* | 61.64 | 58.23 | 59.22 | 53.51 | 61.01 | 62.88 | 48.55 | 62.94 | **66.66** |
| | 5 | 57.26 | 58.17 | 50.62 | 62.20 | 60.80 | 57.45 | 54.36 | 62.89 | 63.82 | *48.82* | 63.07 | **66.61** |
| | 6 | 62.12 | 63.86 | 52.68 | 62.10 | 58.76 | 57.25 | 54.12 | 62.11 | 62.37 | *50.24* | 62.61 | **69.11** |
| | Avg | 60.07 | 59.09 | 49.12 | 62.00 | 59.50 | 58.40 | 54.28 | 62.26 | 63.48 | *48.81* | 63.21 | **68.02** |
| | Std | 1.73 | 2.64 | *5.08* | **0.55** | 1.10 | 0.90 | 1.19 | 0.66 | 1.10 | 1.09 | 0.85 | 1.22 |
| 20 | 1 | 59.81 | 57.52 | *37.81* | 61.86 | 58.07 | 56.82 | 53.73 | 62.75 | 64.41 | 49.80 | 62.29 | **67.07** |
| | 2 | 58.06 | 51.43 | 48.61 | 61.51 | 57.44 | 55.50 | 51.04 | 62.68 | 63.58 | *47.56* | 63.78 | **64.25** |
| | 3 | 61.31 | 55.38 | *42.49* | 62.35 | 56.10 | 57.03 | 54.51 | 61.44 | 64.58 | 49.30 | 64.28 | **68.50** |
| | 4 | 56.09 | 57.82 | *44.68* | 61.61 | 57.23 | 56.77 | 53.12 | 59.39 | 62.33 | 47.65 | 63.43 | **65.25** |
| | 5 | 59.07 | 55.79 | *42.49* | 61.75 | 58.99 | 54.78 | 52.83 | 61.92 | 63.28 | 48.18 | 63.82 | **65.90** |
| | 6 | 59.85 | 55.21 | *43.84* | 61.43 | 58.47 | 56.05 | 53.18 | 61.60 | 62.51 | 50.28 | 62.81 | **66.86** |
| | Avg | 59.03 | 55.52 | *43.32* | 61.75 | 57.72 | 56.16 | 53.07 | 61.63 | 63.45 | 48.80 | 63.40 | **66.31** |
| | Std | 1.79 | 2.29 | *3.52* | **0.33** | 1.02 | 0.89 | 1.16 | 1.22 | 0.94 | 1.16 | 0.73 | 1.50 |
| 40 | 1 | 56.51 | 56.67 | *37.69* | 58.62 | 57.20 | 55.98 | 52.10 | 58.17 | 61.54 | 47.23 | 61.15 | **65.38** |
| | 2 | 54.02 | 54.82 | *38.80* | 57.96 | 55.26 | 52.60 | 50.08 | 58.12 | 61.64 | 44.56 | 60.65 | **64.27** |
| | 3 | 56.53 | 51.22 | *36.93* | 59.18 | 55.30 | 52.12 | 53.85 | 59.63 | 63.27 | 47.98 | 62.10 | **65.36** |
| | 4 | 55.76 | 56.15 | *39.68* | 58.68 | 54.36 | 54.31 | 52.21 | 57.58 | 61.59 | 45.53 | 61.09 | **65.69** |
| | 5 | 57.14 | 52.19 | *41.95* | 57.47 | 56.80 | 50.80 | 49.48 | 57.16 | 61.28 | 44.03 | 60.53 | **61.82** |
| | 6 | 57.06 | 52.31 | *38.13* | 58.36 | 55.44 | 52.50 | 50.18 | 59.40 | 61.01 | 45.00 | 59.08 | **63.33** |
| | Avg | 56.17 | 53.89 | *38.86* | 58.38 | 55.73 | 53.05 | 51.32 | 58.34 | 61.72 | 45.72 | 60.77 | **64.31** |
| | Std | 1.16 | *2.29* | 1.78 | **0.60** | 1.06 | 1.82 | 1.68 | 0.99 | 0.79 | 1.56 | 0.99 | 1.51 |

Table 6: Full results of the label disturbance experiment on **Sleep** data. The **best results** are in bold and we underline the second best results. The *worst* results are denoted in italics.

| r% | Exp | TAS | | | TSC | | | NLL | | | TSC&NLL | | |
|---|---|---|---|---|---|---|---|---|---|---|---|---|---|
| | | MS-TCN2 | AS Former | DiffAct | Mini Rocket | Times Net | Patch TST | SI-GUA | UNI-CON | Sel-CL | SREA | Scale-T | *Con4m* |
| | 1 | 65.86 | 51.41 | 64.82 | 62.11 | *50.25* | 58.51 | 52.09 | 60.64 | 62.57 | 53.09 | 66.80 | **72.26** |
| | 2 | 63.74 | 59.68 | 60.63 | 61.12 | *50.55* | 60.60 | 52.27 | 64.43 | 51.65 | 57.57 | 66.85 | **73.21** |
| - | 3 | 56.03 | 59.04 | 56.41 | 63.92 | *52.17* | 56.22 | 55.21 | 56.51 | 67.27 | 54.99 | 69.28 | **70.52** |
| | Avg | 61.88 | 56.71 | 60.62 | 62.39 | *50.99* | 58.45 | 53.19 | 60.53 | 60.50 | 55.21 | 67.64 | **72.00** |
| | Std | 5.17 | 4.60 | 4.21 | 1.42 | **1.03** | 2.19 | 1.75 | 3.96 | *8.01* | 2.25 | 1.42 | 1.36 |

Table 7: Full results of the label disturbance experiment on **SEEG** data. The **best results** are in bold and we underline the second best results. The *worst* results are denoted in italics.

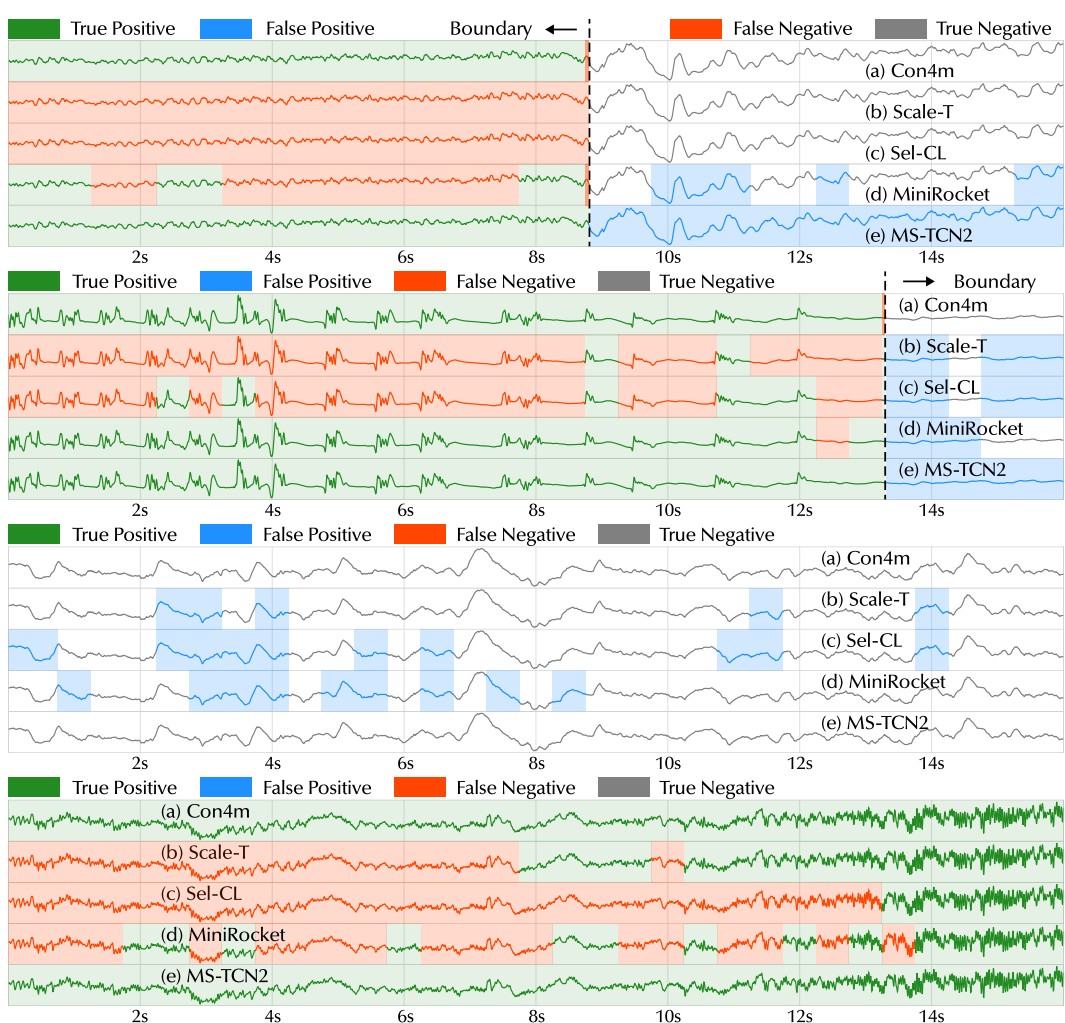

Figure 7: More cases for continuous time intervals in SEEG testing set.

