# OpenReview forum: "Con4m: Context-aware Consistency Learning Framework for Segmented Time Series Classification"
_NeurIPS.cc/2024/Conference — NeurIPS 2024 poster_

### Official Review · Reviewer_Gi9s · 2024-07-09

**Soundness:** 2
**Presentation:** 2
**Contribution:** 2
**Rating:** 5
**Confidence:** 5

**Summary:**

This paper proposes a time-series classification method which exploits temporal consistency (implemented by contextual information). Also, the proposed method can handle noisy class boundaries.

**Strengths:**

S1. This paper presents a novel problem formulation, time-series classification with noisy class boundaries.

S2. The proposed method takes full advantage of temporal consistency, which is an inherent property of time series.

S3. The authors created their own proprietary dataset, SEEG, for extensive experiments.

**Weaknesses:**

W1. First of all, the writing in Introduction and Figure 1 are very misleading. "**Inconsistent** boundary labels" (with "different annotators") and Figure 1(a) indicate that multiple inconsistent annotations coexist for the same time-series instance. However, after reading a few pages, I found a discrepancy between my understanding and the presentation in this paper. Subsequently, I realized that the authors were referring to scenarios in which the class boundaries could be inaccurate. That is, there is **one** possibly erroneous boundary label for each transition. This presentation issue really affected my negative impression on this paper.

W2. I do not agree that existing studies largely focus on the assumption of i.i.d. It is very easy to find the existing work that considers temporal consistency. For example, please see https://cs.stanford.edu/people/jure/pubs/ticc-kdd17.pdf. More recently, please see https://openreview.net/pdf?id=gjNcH0hj0LM. That is, many existing methods consider that consecutive timestamps tend to have the same class label.

W3. Con-Attention is the same as Anomaly-Attention. Please see Equation (2) in https://arxiv.org/pdf/2110.02642. Therefore, along with W2, the technical novelty for the first contribution is not very significant.

W4. Section 2 is not tightly connected to subsequent sections. In my opinion, Theorem 2 is too obvious, and it does not need to be formulated as a theorem. In fact, it does not look like a formal theorem.

W5. The Tanh function fitting seems to be needed for each segment boundary and is not straightforward (involved with many parameters). Thus, the training efficiency could be an issue. It would be good to report the training cost of the proposed method.

W6. The authors assume that the center of a class interval has the highest confidence. This assumption could be true or false depending on the domain. As in the motivating example, if the annotators have a difficulty in identifying the end times of seizure waves, the highest confidence will appear earlier than the center.

W7. The authors made several strong assumptions or heuristics. For example, it is assumed that consecutive segments almost span at most two classes; a class interval is divided into five levels; a training cap is given as five epochs. Thus, this paper lacks rigorousness and generalizability.

W8. It is not trivial to set up the hyperparameter value, especially, $E_\eta$. The default value is not guaranteed to work well for other datasets. A detailed guideline is necessary.

W9. The degree of noisy labels in each dataset should be analyzed in detail. First, it is not clear why there is no noise in fNIRS and Sleep, while there is real noise in SEEG. Is the annotation strategy different in these datasets? How do you know the ground-truth labels in SEEG? Second, can you measure the degree of real noise just like the noise rate in the image classification domain? Can you approximate the corresponding value of $r$ for this degree of noisy labels in SEEG? Overall, without detailed information on noisy labels, it would be difficult to analyze the effect of noisy label learning.

---

I have adjusted my rating to 5 during the discussion period.

**Questions:**

See W1~W9.

**Limitations:**

In my opinion, the generalizability of the proposed method is not clearly verified. In this regard, the authors did not properly discuss the limitations of the proposed method. Please see my comments to further improve the manuscript.

---

> ### Author Rebuttal · Authors · 2024-08-07
>
> W1. We apologize for any confusion caused. Fig. 1 is intended to emphasize that, despite the similarity in seizure patterns, different annotators can still provide inconsistent labels for the same type of seizure and across different recordings from the same patient.  For simplicity, we only illustrated a segment of brain signals to represent a seizure pattern type, rather than a single instance. This will be explicitly stated in the revised version. Additionally, inconsistent labels do not equate to erroneous labels, as there is a lack of unified quantitative metrics to define boundaries between different classes.
>
> ---
>
> W2. Our focus is on supervised TSC, which assigns each segment a single label, as stated in the problem definition in Sec. 3. We acknowledge that the studies you referenced consider temporal consistency. However, the first paper focuses on clustering, which is fundamentally distinct because it involves setting/learning the number of clusters within a sequence. The second paper concentrates on active learning. While these studies leverage temporal consistency, they do not align directly with the supervised TSC models we discuss. And our work explicitly integrates temporal consistency into the modeling of supervised TSC from both the data and label perspectives.
>
> ---
>
> W3. The only similarity between the two is the parallel computation of $G^l$. Anomaly Transformer uses the similarity between inter-sequence and intra-sequence distributions to measure anomaly scores, whereas our approach aims to achieve smoother representations through fusion. We have cited this work in the main text (line 143) and have not referred to its code.
>
> ---
>
> W4. While Theorem 2 may seem intuitively obvious, we provide a more formal proof from an information-theoretic perspective (lines 109-121). Based on this theorem, we designed our model from both the data and label perspectives, as detailed in Sec. 3.1 (lines 142-149) and 3.2 (lines 170-173, 182-187).
>
> ---
>
> W5. Assuming the number of consecutive input time segments is $L$, the hidden representation dimension is $D$, the number of classes is $C$, and the internal iteration steps of the Tanh function fitting are $I$, the time complexity of the function fitting module is $\mathcal{O}(ICL)$. The complexity of the encoder’s final prediction layer is $\mathcal{O}(DCL)$. In practice, keeping $I < D$ helps manage the time complexity of the function fitting module. In our experiments, we set $I=100$ and $D=128$, with average training times reported per epoch.
>
> The additional computational cost is higher for the Sleep dataset due to its classification as a five-class problem. SEEG, like fNIRS, is a binary classification problem. However, the additional overhead from function fitting is relatively smaller due to the larger input sample window length and the total number of training samples. As a result, the function fitting module has a higher computational overhead on datasets with more classes, which is a limitation of our model design. However, this overhead becomes less significant with larger datasets.
>
> |             | fNIRS        | Sleep        | SEEG         |
> | ----------- | ------------ | ------------ | ------------ |
> | Con4m       | 11.45s       | 37.73s       | 24.75s       |
> | w/o fitting | 1.71s        | 3.92s        | 6.78s        |
> |             | $\times$6.70 | $\times$9.63 | $\times$3.65 |
>
> ---
>
> W6. Our assumption is that annotators “tend to reach an agreement on the most significant core part of each class,” which is common in temporal action segmentation. We apologize for any confusion caused by Fig. 1; we used real data, so the illustration does not depict an entire seizure episode due to its length. In practice, doctors often disagree on both the exact start and end times of seizures. While domain knowledge can aid in sample selection, our design remains general when such prior knowledge is unavailable.
>
> ---
>
> W7. We acknowledge in lines 334-335 that modeling transitions for a single class is a limitation. However, based on domain knowledge and statistical information from the datasets, we can select an appropriate window length that meets this condition. For the auxiliary parameters $N^l$ and $E_g$, we do not consider them core hyperparameters of the model. We ensure that the model’s loss approaches convergence for the new level data within $E_g$ epochs. See more details in global rebuttal G2.
>
> ---
>
> W8. Although we tuned $E_\eta$ on the SEEG dataset, we used the same hyperparameter across all 4 datasets, demonstrating Con4m’s robustness. According to Appendix C, $E_\eta$ should not be set too low. For practical use, start with a value of 10 and adjust in intervals of 10 until the validation set performance peaks or slightly declines.
>
> ---
>
> W9. The fNIRS and Sleep datasets are publicly available with curated labels, so they are considered noise-free. In contrast, SEEG data is derived from real clinical datasets and annotated by multiple experts, resulting in naturally inconsistent labels. We employ a voting mechanism (Appendix G, lines 746-749) to minimize discrepancies in test labels. However, due to the lack of a unified standard, inconsistencies in SEEG do not imply “incorrect” labels but rather reflect differences in experiences. Unlike video segmentation, where single-frame semantics are clearer, SEEG annotations can vary significantly.
>
> Our goal is to harmonize, not correct, these labels using methods inspired by noisy label learning. Therefore, we cannot directly calculate an r value for SEEG but instead use indirect label substitution experiments (Sec. 4.3) to validate Con4m’s effectiveness in label harmonization. Comparisons with other noisy label learning baselines and ablation studies further demonstrate Con4m’s ability to handle inconsistent labels.

---

> ### Comment · Reviewer_Gi9s · 2024-08-09
> **Will keep my current rating**
>
> Thank you for your responses. I have carefully read the authors' rebuttal.
>
> (1) As I mentioned earlier, the overall presentation is somewhat confusing. I now better understand the authors' original intention. However, the current draft needs a major rewrite. The differences between erroneous and inconsistent boundary labels are still not clear to me. To my understanding, this paper deals with possibly erroneous boundary labels caused by inconsistent annotations. Because we cannot review the revised draft, it would be hard to increase the current rating.
>
> (2) The responses on the novelty issues (W2 and W3) and the generalizability issues (W7) are not very convincing to me.
>
> (3) The in-depth analysis w.r.t. the degree of noise or inconsistency (W9) would be necessary to further strengthen this work. A voting mechanism applied to SEEG means curation? If you apply the voting mechanism to the training set of SEEG, it becomes the same status as fNIRS and Sleep? There are several unclear points regarding the motivation and experiment setting.
>
> Overall, I would like to keep my current rating.

---

> > ### Author Response · Authors · 2024-08-12
> >
> > Thank you for your response. We would like to provide further clarification.
> >
> > (1) This involves defining what constitutes "erroneous." Erroneous labels imply that we assume there exists an objectively true label behind each label. However, inconsistent labels simply represent differences in annotator experience. As discussed in lines 50-56 of the main text, for MVD, the boundaries between states are not clearly defined, or the transitional states themselves represent a mixed state. Consequently, behind inconsistent labels, there is no artificially defined true label. Perhaps a model could assist in defining this true label, but thus far, our work aims to harmonize this inconsistency as much as possible to reduce the instability of model training and enhance its performance.
> >
> > (2) The two works you mentioned in W2, one designed for clustering tasks and the other for active learning, both fall outside the scope of TSC. Our design for representing continuity and label coherence is two sides of the same coin, closely aligned with our intent and motivation. We conduct hyperparameter analysis for W7 in G2, and in Section 5, we spell out the limitation regarding the diversity of class transition behaviors.
> >
> > (3) As mentioned in (1), there is no objectively quantifiable true labels behind inconsistent labels, making it challenging to quantify inconsistency. The voting mechanism refers to bringing multiple experts together to collectively decide the boundaries of the test set, rather than independently annotating different patients or files. However, this approach incurs significant costs, as each patient's record is extensive (spanning several days or even a dozen days), hence we can only apply it to the test set. If the training set were also handled in this manner, we have reason to believe that SEEG data could be considered equivalent to clean datasets like fNIRS and Sleep.

---

> > > ### Comment · Reviewer_Gi9s · 2024-08-13
> > > **Thanks for your clarification**
> > >
> > > I now have a better understanding on the problem setting and experiment configuration. The overall presentation should be significantly improved to avoid any confusion on the problem setting and experiment configuration. In particular, Figure 1 must be revised to properly explain the motivation. I will adjust my rating to 5, assuming that the authors will reflect my comments. Thank you very much.

---

> > > > ### Author Response · Authors · 2024-08-13
> > > > **Thank you**
> > > >
> > > > Thank you for your trust and raising the score.
> > > >
> > > > (1) Regarding the problem setup, we will emphasize the meaning of inconsistent labels starting from line 52: "However, due to inherent ambiguity and a lack of unified quantification standards, for MVD, the boundaries between states are not clearly defined, or the transitional states themselves represent a mixed state. Consequently, behind inconsistent labels, there is no artificially defined true label. Therefore, our work aims to harmonize this inconsistency as much as possible to reduce the instability of model training and enhance its performance."
> > > >
> > > > (2) Regarding Figure 1, our current plan for modification is as follows: "Place (c) at the top and elongate it, then vertically superimpose two other distinct but similar types of seizure waves. The annotations from three physicians will be marked on these three brain signals. The labels lv.1-lv.5 will be placed inside the signals to reduce visual clutter. The three signals collectively point to the x-axis of figure (d)."
> > > >
> > > > (3) Regarding the experimental setup, we will insert the content about SEEG from Appendix G starting from line 254: "Notice that SEEG data is derived from real clinical datasets and annotated by multiple experts, resulting in naturally inconsistent labels. We employ a voting mechanism which brings annotators together to collectively decide the boundaries to minimize discrepancies in test labels. Considering the high cost of this approach, we do not apply it to the training and validation sets. Therefore, we leave the test group aside and only change the training and validation groups to conduct cross-validation."

---

### Official Review · Reviewer_uhXL · 2024-07-12

**Soundness:** 3
**Presentation:** 4
**Contribution:** 3
**Rating:** 6
**Confidence:** 4

**Summary:**

This paper addresses the challenge of segmented time series classification (TSC) for Multiple classes with Varying Duration (MVD) data. The authors propose Con4m, a consistency learning framework that leverages contextual information with a focus on inconsistent boundary labels. The method incorporates continuous contextual representation encoding, context-aware coherent class prediction, and a label consistency training framework. Experiments on 3 datasets demonstrate Con4m's superior performance compared to state-of-the-art baselines, especially in handling inconsistent labels.

**Strengths:**

- Originality: The paper introduces a new approach to segmented TSC for MVD data, addressing challenges often overlooked in existing TSC models. The Con4m framework creatively combines ideas from curriculum learning, noisy label learning, and temporal action segmentation.

- Quality: The theoretical analysis in Section 2 and throughout provides a solid foundation for the proposed method. The experiments are comprehensive, including comparisons with various baselines, label disturbance experiments, and ablation studies.

- Clarity: The paper is well-structured and clearly written. The figures, especially Figure 2 and Figure 3, effectively illustrate the proposed method's architecture and workflow.

- Significance: The work has potential implications for various domains dealing with MVD data, such as healthcare and activity recognition. The label harmonization approach could be particularly valuable in scenarios where obtaining consistent labels is challenging or costly.

**Weaknesses:**

- Limited exploration of hyperparameters: The paper doesn't thoroughly discuss the sensitivity of the method to key hyperparameters, such as N_l = 5 in the label consistency framework or E_g = 5 for curriculum learning.

- Scalability concerns: The paper doesn't address how the method would scale to larger datasets or longer time series. This could be a limitation for practical applications with high-dimensional or long-duration data.

- Comparison with semi-supervised methods: Given that the method deals with inconsistent labels, it might be beneficial to compare it with semi-supervised learning approaches that are designed to handle partially labeled or noisy data.

- Generalization to other domains: While the method is tested on three datasets, they are all from the healthcare domain (all 3 of them focusing on brain signal analysis). It would be valuable to see how well the approach generalizes to other domains with MVD data, such as activity recognition or financial time series. For example, initiatives like WOODS (https://arxiv.org/abs/2203.09978/) offer diverse timeseries data for evaluations.

**Questions:**

1. In the label consistency framework, why is N_l = 5? Was this value determined through hyperparameter search? How sensitive is the method to this choice?

2. Why was fNIRS not used in the ablation studies? Is there a specific reason for excluding this dataset from the detailed analysis?

3. How does the computational complexity of Con4m compare to the baseline methods? Is there a significant increase in training time or memory requirements?

4. Have you explored the potential of applying Con4m to semi-supervised or few-shot learning scenarios, where only a portion of the data has reliable labels?

5.  The paper mentions that Con4m modifies approximately 10% of the training labels in the SEEG data. How does this percentage vary across different datasets or disturbance ratios? Is there a way to estimate the optimal percentage of labels that should be modified?

6.  Could you provide more insights into the choice of the hyperbolic tangent function for prediction behavior constraint? Have you experimented with other monotonic functions, and if so, how do they compare?

**Limitations:**

See the questions above. I would especially emphasize the narrow benchmarks and tasks. Learning across segments is a broader problem within the timeseries domain.

---

> ### Author Rebuttal · Authors · 2024-08-07
>
> W1&Q1. We consider $N_l$ and $E_g$ as auxiliary hyperparameters that work together rather than core hyperparameters of the model. When selecting these values, we focus on ensuring that the model’s loss approaches convergence for the newly added level within $E_g$ epochs. In contrast, $E_\eta$ has a more significant impact on the model’s performance. See more details in global rebuttal G2.
>
> ---
>
> W2. From an architectural perspective, Con4m is built on the Transformer framework, allowing for scalability by adjusting the hidden layer dimensions or stacking layers. From a data perspective, time series data differs from NLP sequences as it consists of continuous numerical values rather than discrete tokens. By adjusting the duration of each time segment (patch), we can control the sequence length L. Consequently, Con4m can be scaled to handle longer time series by appropriately tuning the patch length, the number of CNN layers, and the size of the convolutional kernels.
>
> ---
>
> W3&Q4. Our method differs from semi-supervised TSC methods that assume reliable labels on partially labeled datasets. In contrast, we begin with initial labels for the entire dataset, but their reliability remains unclear. Our model is specifically designed to evaluate label reliability as part of its learning process, addressing inconsistencies not typically covered by semi-supervised methods. Therefore, we cannot consider semi-supervised or few-shot methods as a baseline or apply them to our task due to these differences in task settings.
>
> ---
>
> W4. Thank you for your valuable suggestion. Based on the link you provided for WOODS, we identified that the Human Activity Recognition (HHAR) dataset aligns with our experimental setup to some extent. Therefore, we incorporated the HHAR dataset in our experiments. Please refer to the global rebuttal G1 for more details.
>
> ---
>
> Q2. Table 2 demonstrates that fNIRS has relatively clear boundaries, as TAS models perform well on this dataset even without considering label consistency. Therefore, to more effectively highlight Con4m’s advantages in handling inconsistent labels, we decided not to include the fNIRS dataset in the ablation studies.
>
> ---
>
> Q3. We believe that due to the significant differences in architecture and domain among various baselines, directly comparing computational complexity would be unfair. For example, models based on the Transformer architecture have a significantly higher time complexity than those based on CNNs. Therefore, we only compare the time complexity of Con4m with that of a standard Transformer.
>
> The time complexity of Con4m can be divided into two main parts. The primary computational cost is on the same order as vanilla Transformer. Assuming the number of consecutive input time segments is $L$, the hidden representation dimension is $D$, the number of classes in the classification task is $C$, and the local iteration count of the function fitting module is $I$.
>
> - Con-Transformer: The time complexity of vanilla self-attention is $\mathcal{O}(LD^2+L^2D)$, and the time complexity of the Gaussian kernel branch is $\mathcal{O}(LD^2+L^2)$.
> - Coherent class prediction: The time complexity of Neighbor Class Consistency Discrimination is $\mathcal{O}(LDC)$, and the time complexity of the Tanh function fitting is $\mathcal{O}(ICL)$.
>
> Therefore, the computational cost and bottlenecks of Con4m are similar to vanilla Transformer.
>
> ---
>
> Q5. Since label inconsistency is difficult to precisely define, it is challenging to determine an optimal percentage of label modifications. However, during training, one of our criteria for stopping is when the model no longer changes additional labels. You can find this implementation in our code. Generally, datasets with higher noise levels tend to have a higher percentage of label changes. For example, the fNIRS-0 dataset has a change rate of 3%, while SEEG has a change rate of 13%. Similarly, for the Sleep dataset with disturbance ratios of 0%, 20%, and 40%, the change rates are 10%, 12%, and 18%, respectively.
>
> ---
>
> Q6. The Tanh function is particularly well-suited to encapsulate the four behaviors depicted in Figure 3. As a widely used activation function, it is easier to understand and manipulate. In contrast, functions like arctangent and arcsine are more complex and harder to control. Appendix B demonstrates Tanh’s ease of optimization and its precise fitting capabilities.

---

> > ### Comment · Reviewer_uhXL · 2024-08-12
> >
> > I want to thank the authors for their thorough responses. I have updated my score accordingly.

---

> > > ### Author Response · Authors · 2024-08-12
> > > **Thank you**
> > >
> > > Thank you very much for raising your score, which is highly valuable to us. We have gained a lot from your comments, such as identifying the HHAR dataset from the WOODS benchmark as one of our test datasets. If you have any further concerns or suggestions, please do not hesitate to share them with us.

---

### Official Review · Reviewer_7tXB · 2024-07-12

**Soundness:** 3
**Presentation:** 3
**Contribution:** 3
**Rating:** 5
**Confidence:** 4

**Summary:**

The paper proposes Con4m, a novel framework for time-series classification and temporal action segmentation that leverages contextual information. The framework is designed to improve the prediction accuracy by incorporating context from surrounding data segments.
The proposed method combines time-series classification (TSC) and temporal action segmentation (TAS) in a unified framework, making it versatile for different types of sequential data. It tries to address the issue of common time series classification methods which overlook the dependencies between consecutive segments.
The method demonstrates robustness to label disturbances, showing significant improvements over baseline models across different datasets (fNIRS, Sleep, and SEEG).

**Strengths:**

-	The paper proposes context-aware boundary detection and classification for long multi-class time series.  The topic is very interesting and focuses on a real challenge in processing time-series data.
- Con4m integrates contextual information to improve the coherence and accuracy of predictions. This approach helps in better recognizing boundaries and transitions in time-series data.
-	The paper is well-written and presented with several example illustrations.
-	The model is Compared against different time-series classification baselines. The evaluation shows great performance across Sleep and SEEG datasets and demonstrates robustness to label disturbances

**Weaknesses:**

1.	What if combining other models with a change-detection model to detect segment boundaries first and then classify the windows within the segments? That would be great to see the impact of Con4m in comparison with other baselines in this setup.
2.	Since Con4m is capable of doing segmentation and classification together, it will be necessary to evaluate and compare the impact of each with other SOTA.
Minor change:
-	Figure 4(b) has been referred to before Figure 4(a). Please fix that
3. The experiments are limited, and somewhat unfair. The chosen baselines do not claim ability in segment detection. There are several works that have done segment or change point detection, and they need to be included in baselines. Also, evaluations are limited to three datasets.
-	Minor suggestion: It would be nice to visualize the whole framework to show each module.

**Questions:**

See the initial two comments (no 1 and no 2) in weaknesses. In addition:

3.	In Label Disturbance Experiments: It is not clear how the authors evaluated the models. Is it based on the new disturbed boundaries or the original boundaries? What is the evaluation setup for the other baselines as they are not considering the labelled boundaries in their calculations (I assume they should be boundary-agnostic).
4.	While section 4.3 shows an interesting set of experiments, I believe it only measures the degree of agreement between the baseline models and Con4m about the modified labels. Unless authors can ensure all the labels modified by Con4m are correct and accurate

**Limitations:**

The following limitations are not considered/discussed:

-	Although I appreciate the novelty of the method, the final improvement is subtle compared to other baselines given they have not claimed any ability in segment detection. Also, the evaluations are limited to 3 datasets.
-	As the authors mentioned in the conclusion the model heavily relies on the availability of labels even for the segmentation part. While most segmentation models are unsupervised, I suggest including comparison by combining other baselines with unsupervised segmentation methods.

---

> ### Author Rebuttal · Authors · 2024-08-07
>
> W1&2.
>
> 1. Our setup differs significantly from segmentation models (FLOSS[1], ESPRESSO[2], ClaSP[3]) in that they are able to identify change points but are unable to determine the specific classes before and after these points, particularly in multi-class tasks.
> 2. Our public datasets are multivariate, which presents a challenge for the performance of most segmentation models (ClaSP[3]), which are designed to handle univariate time series.
> 3. Our use case may be impractical and complex due to the necessity of setting/learning  parameters, including the number of segments and the length of subsequences, in segmentation models (FLOSS[1], ESPRESSO[2]).
>
> Nevertheless, we conduct an exploratory analysis of ClaSP on the SEEG dataset (the only univariant dataset) and present the results with two metrics for reference. These results are solely intended for illustration purposes, as unsupervised segmentation models are not well-suited to our scenario and setup.
>
> [1] Gharghabi, Shaghayegh, et al. "Domain agnostic online semantic segmentation for multi-dimensional time series." (2019).
>
> [2] Deldari, Shohreh, et al. "Espresso: Entropy and shape aware time-series segmentation for processing heterogeneous sensor data." (2020).
>
> [3] Ermshaus, Arik, Patrick Schäfer, and Ulf Leser. "ClaSP: parameter-free time series segmentation." (2023).
>
> ---
>
> W3. Thank you for your valuable questions and suggestions. We will reorder Figures 4(a) and 4(b) and attempt to combine Figures 2 and 3 into a single illustration. However, we would like to emphasize that Con4m was not designed specifically for segmentation. Our modeling is consistently based on Problem Definition 3.1, which involves making independent predictions for each segment and then integrating them into coherent predictions. Our work aims to highlight a point often overlooked by current mainstream supervised TSC approaches: the temporal dependency and coherence of contextually classified samples. Experiment 4.3 further illustrates the contributions of our work.
>
> On the other hand, all the baselines related to temporal action segmentation (TAS) are equipped with segmentation capabilities, as they are specifically designed for segmentation tasks. Since Con4m is a supervised learning model, we selected representative supervised learning models in TAS for comparison. Therefore, the selection and comparison of baselines are fair. Nevertheless, we have considered your recommendation to incorporate unsupervised segmentation models for comparison in W1&2.
>
> ---
>
> Q3. Our experimental setup is consistently aligned with Problem Definition 3.1 (lines 136-140). All models are trained on the disturbed data and tested on the original noise-free data. We introduce disturbances at the timestamp level of the original data (lines 236-243), then sample time intervals based on these disturbances and segment them. All models treat each time segment as an instance for prediction and evaluation, which aligns with the TSC task setup. The evaluation setup itself is independent of whether a model can detect boundaries. Con4m does not explicitly perform boundary detection, whereas TAS models are specifically designed for boundary segmentation.
>
> ---
>
> Q4. Section 4.3 effectively demonstrates Con4m’s prediction consistency. Moreover, Fig.5 and Fig.8 use color coding to visually highlight the alignment between the model’s predictions and the actual labels. It is evident that, unlike Con4m, other models fail to provide predictions that are both coherent and accurate. In practical applications, Con4m offers substantial improvements in usability, significantly outperforming other models in overall classification metrics. Dispersed and inconsistent predictions can hinder doctors from quickly identifying seizure regions without examining large amounts of raw data.

---

> > ### Author Response · Authors · 2024-08-09
> > **Results for unsupervised segmentation model ClaSP**
> >
> > Apologies for the delayed results. For the F1 score provided by ClaSP, we input each time interval into ClaSP to obtain a score, and then average the scores of all time intervals. Additionally, we transform Con4m's predicted results from time segments to timestamps and use them to calculate the F1 score proposed by ClaSP. The specific scores are as follows: ClaSP: 0.854; Con4m: 0.930. The comparison of these results is quite fair.
> >
> > To align with our evaluation metrics, we assign all possible class combinations to the timestamps on either side of the change points detected by ClaSP. We then select the combination with the greatest overlap with the true test labels for ClaSP. Finally, the same segmentation and evaluation processes are executed to obtain the results. The specific f1 scores are as follows: ClaSP: 0.824; Con4m: 0.720. While this approach provides a test result, it clearly involves test label leakage, rendering it unfair and unsuitable for real-world applications.

---

> > ### Comment · Reviewer_7tXB · 2024-08-12
> >
> > I would like to thank the authors for responding to my questions. The provided clarifications for Q3 and Q4 makes the evaluation setup more clear.

---

> > > ### Author Response · Authors · 2024-08-12
> > > **Further discussion**
> > >
> > > Thank you for your response. We have also included the analysis of unsupervised segmentation models in our setup, along with the configuration and comparative results of ClaSP on SEEG data. Do you have any further questions or concerns regarding these responses? We are more than willing to engage in further discussion to enhance the quality and score of our work.

---

> > > ### Author Response · Authors · 2024-08-13
> > > **Further discussion**
> > >
> > > We would like to reiterate the connection and distinction between unsupervised segmentation tasks and our scenario:
> > >
> > > (1) Pure segmentation models can only identify the positions of change points, without specifying the exact classes on either side of the change points, especially in multi-class scenarios. Although Con4m was not specifically designed for segmentation tasks, it can still transform from segment prediction to point prediction and provide segmentation results. Therefore, based on the SEEG dataset, we evaluate Con4m fairly according to the F1 scores provided by ClaSP. Experimental results are as follows: ClaSP: 0.854; Con4m: 0.930. We believe this result corroborates the conclusion drawn in Section 4.5.
> > >
> > > (2) Temporal Action Segmentation (TAS) models are specifically designed for video segmentation. Following your indication of being "capable of doing segmentation and classification together," the TAS models perfectly fulfill both criteria as they can provide a specific class prediction for each video frame/time segment. Hence, our choice of baseline models is comprehensive and fair.
> > >
> > > (3) Many segmentation models are primarily designed for univariate time series and face challenges when handling multivariate time series data. Therefore, we could only compare ClaSP on the unique univariate dataset SEEG.
> > >
> > > As we are not experts in the field of unsupervised segmentation, we welcome any corrections if there are any inaccuracies or omissions. Additionally, we would greatly appreciate your timely communication and sharing of thoughts that integrate such models into our scenario.

---

> > > > ### Comment · Reviewer_7tXB · 2024-08-13
> > > >
> > > > The authors have added a comparison with an unsupervised segmentation method. However, the comparison is only done across one of the datasets out of three.
> > > > Overall, the authors' approach is a combination of segmentation and classification, both are fully supervised.
> > > > These limitations need to be addressed and clarified in the paper, as there are existing approaches for change point detection that are self-supervised.
> > > >
> > > > I also agree with the last reviewer's comment. The authors need to improve the work by explaining the experiment setup more clearly.
> > > >
> > > > Given the edits and discussions, I am happy to increase the score to 5.

---

> > > > > ### Author Response · Authors · 2024-08-14
> > > > > **Thank you**
> > > > >
> > > > > Thank you for your response and score improvement.
> > > > >
> > > > > As ClaSP is only capable of handling univariate time series data, we conduct experiments solely on the SEEG dataset, as the other three publicly available datasets consist of multivariate time series. We are still perplexed as to why Con4m should be compared with unsupervised segmentation models. From our perspective, Con4m must rely on independent sample predictions $\hat{p}$ and contextual predictions $\tilde{p}$ to perform function fitting to determine the class boundary. However, unsupervised segmentation models cannot specify the exact classes of the samples, while the Temporal Action Segmentation (TAS) models can achieve this. Therefore, we provide an exposition (lines 43-49) and comparison of the TAS models. Nevertheless, we concur that this comparative result can be utilized for the quantitative metrics in Section 4.5 to highlight Con4m's performance in segmentation precision and coherence. Consequently, we will include ClaSP's F1 scores in Figure 5. Additionally, at line 334, we will insert: "Con4m is a combination of segmentation and classification, both of which are fully supervised. Exploring its application in unsupervised segmentation tasks is worthwhile." Regarding the experimental setup, we will supplement it in accordance with the point (3) in the response (https://openreview.net/forum?id=jCPufQaHvb&noteId=fNo5b5RBTi) with Reviewer Gi9s.

---

### Official Review · Reviewer_gPgT · 2024-07-12

**Soundness:** 3
**Presentation:** 3
**Contribution:** 4
**Rating:** 8
**Confidence:** 4

**Summary:**

The authors proposed a learning framework called $\textit{Con}4\textit{n}$ that leverages contextual prior of Multiple classes with Varying Duration (MVD) to enhance the discriminative power of consecutive time series segments while harmonizing inconsistent labels associate to these later. The authors stated and shown through extensive experiment that their framework enables to encode underlying temporal dependence between consecutive time series segments and mitigate the data annotation error that may be due to discrepancy of expertise.

**Strengths:**

- Originality: The authors propose a sophisticated architecture for encoding the temporal dependency between successive time series segments during a classification task. The originality of this work lies in the fact that the classification process is reinforced by a prior contextual information encoded by a Gaussian kernel. Their approach is more realistic than previous work, which assumes an independent and identical distribution of successive time series segments.

- Quality: The article is well-written and structured. The experiment is consistent enough;

- Clarity:  Although the document is fairly clear overall, there are still a number of areas for improvement (please refer to weaknesses and questions);

- Significance: The proposed solution, once mature, will undoubtedly benefit many domains in which time series classification can be very challenging due to the complexity of data structure and divergence of experiences.

**Weaknesses:**

- Figure 2 is not intuitive enough to understand the overall architecture of the model;

- $\hat{p}$ and $\tilde{p}$ are mentioned without explanation. Even though for $\hat{p}$  it is quite obvious, it must be explained clearly (see paragraph $\textbf{Neighbor Class Consistency Discrimination}$.

- Space and Time complexity: Although efficient, the proposed model may require significant memory and computational time in large-scale applications.

- The model may fail when confronted with seasonal time series data. This is because the Gaussian kernel used is not suitable for capturing the periodic pattern of the data;

- Execution time is not included in the results. This is an important measure to report, as it plays a crucial role in sensitive areas such as medicine, where accuracy and prediction time are just as important.

**Questions:**

- What features do you use in positional encoding? Is it the order index of time series segments or timestamps? Please mention it in the main text;

- Can the model work with multivariate time series data, i.e. using the observation of various features in a time segment to perform a classification task? If so, how will you encode the multivariate time series segments?

- Do the authors think that their model will still work correctly when faced with seasonal time series data, given that the Gaussian Kernel used is not appropriate for capturing the periodic pattern of the data?

- Why did the authors choose the attention mechanism to merge $z_s^l$ and $z_g^l$? Although probably less efficient, have they tried simple techniques such as concatenation, Hadamard product or addition, which have the advantage of reducing spatial and temporal complexity?

**Limitations:**

Although the authors have discussed the limitations of their work, the space and time complexity of the model is an additional limitation. Furthermore, unless the reviewer has missed some points, it appears that the proposed model has only been evaluated on univariate time series data (please correct the reviewer if he is wrong). The reviewer suggests investigating these points in future work.

---

> ### Author Rebuttal · Authors · 2024-08-07
>
> Thank you very much for recognizing our work.
>
> W1. We will integrate Figures 2 and 3 into a single illustration to improve clarity.
>
> ---
>
> W2. We will explicitly clarify the meanings of `p^` and `p~` in the main text: `p^` represents the model's independent prediction for a sample, while `p~` denotes the context-aware prediction that incorporates the results from neighboring samples.
>
> ---
>
> W3. The time complexity of Con4m can be divided into two main components. The primary computational cost is on the same order as vanilla Transformer. Assuming the number of consecutive input time segments is $L$, the hidden representation dimension is $D$, the number of classes in the classification task is $C$, and the local iteration count of the function fitting module is $I$.
>
> - Con-Transformer: The time complexity of vanilla self-attention is $\mathcal{O}(LD^2+L^2D)$, and the time complexity of the Gaussian kernel branch is $\mathcal{O}(LD^2+L^2)$.
> - Coherent class prediction: The time complexity of Neighbor Class Consistency Discrimination is $\mathcal{O}(LDC)$, and the time complexity of the Tanh function fitting is $\mathcal{O}(ICL)$.
>
> Therefore, the computational cost and bottlenecks of Con4m are similar to vanilla Transformer.
>
> ---
>
> W4&Q3. Thank you for the insightful question. Con4m is primarily designed for classification tasks, focusing on extracting structures and patterns within time segments. When dealing with datasets that have periodic patterns, we believe Con4m can still effectively capture these periodicities, provided that a complete cycle is included within a single classification instance. In this context, the Gaussian kernel acts as a one-dimensional Gaussian smoothing filter. We believe that classification tasks are typically less sensitive to periodic patterns compared to forecasting tasks. Furthermore, we have included a human activity recognition dataset HHAR, where sensor data naturally exhibit periodic changes as people walk or run. Con4m performs well on the six-class HHAR dataset, demonstrating its ability to capture periodic patterns.
>
> ---
>
> W5. The execution time of Con4m on SEEG data is as follows: (1) Offline training (124,000 training samples + 62,000 validation samples): approximately 30s/epoch; (2) Batch inference (batch size of 64 with 62,000 test samples): approximately 14s. These times fully meet the practical requirements for inference.
>
> ---
>
> Q1. We employ learnable absolute positional encoding. The absolute positional encoding is equivalent to the order index of time series segments, as the fundamental unit of our model is a time segment. This clarification will be addressed in the primary text of the paper.
>
> ---
>
> Q2. Con4m is capable of handling multivariate time series data. As shown in Table 1, the fNIRS and Sleep datasets have 8 and 2 features, respectively. We employ a multi-channel CNN to jointly extract information from these features (see Figure 2), as the primary focus of this paper is not on multi-channel modeling. However, Con4m can be seamlessly integrated with existing multi-channel modeling approaches by using their output representations as inputs to Con4m.
>
> ---
>
> Q4. Summation is the most direct and straightforward approach to achieving smoother representations, as concatenation and the Hadamard product do not explicitly provide this effect. Additionally, we enable the model to adaptively learn the necessary degree of smoothness, as sample representations in transition regions should not be continuous with those from different classes.

---

> > ### Comment · Reviewer_gPgT · 2024-08-10
> > **Satisfied with the answers.**
> >
> > Dear authors, thank you for your detailed responses. My concerns were clarified.

---

> > > ### Author Response · Authors · 2024-08-12
> > > **Thank you**
> > >
> > > Thank you once again for your recognition and affirmation of our work. If you have any further questions or concerns, please feel free to let us know at any time.

---

### Author Rebuttal · Authors · 2024-08-07

G1 (Results for a new dataset).

Based on the suggestion from reviewer uhXL, we conducted a search of the WOODS dataset and found that the Human Activity Recognition (HHAR) subset aligns well with our scenario and setup. Consequently, we include the HHAR dataset in our experiments. Due to time constraints, we only report the results of the best-performing model from each category of baselines, and we plan to include the remaining results in the revised version.

We follow the preprocessing steps outlined in WOODS (https://arxiv.org/pdf/2203.09978), dividing the HHAR dataset according to the device. To ensure balanced samples, we combined data from the Galaxy S3 Mini, LG watch, and Gear watch into a single group. A 6-fold cross-validation experiment is conducted in accordance with the Sleep dataset's configuration. As Table 1 in the rebuttal pdf shows, Con4m still achieves competetive or superior performance in HHAR dataset. Due to the distribution discrepancies among devices, there are significant variations in experimental results across different groups.

---

G2 (Results for hyperparameter analysis).

We consider $N_l$ and $E_g$ as auxiliary hyperparameters that work in tandem rather than core hyperparameters of the model. When selecting these values, we empirically observe that the model’s loss converges for the newly added levels within $E_g$ epochs.

As shown in Figure 1 in the rebuttal pdf, smaller values for either $N_l$ or $E_g$ can hinder the model’s ability to fit data at newly introduced levels, leading to a decline in performance. However, when $N_l$ and $E_g$ are set within the range of 4 to 7, Con4m demonstrates stable performance, indicating that the model is not particularly sensitive to these hyperparameters. In contrast, the parameter $E_\eta$ has a more significant impact on the model’s performance, and careful tuning of this hyperparameter is more crucial.

---

### Decision · Program_Chairs · 2024-09-25

**Decision:**

Accept (poster)

**Comment:**

This paper should be accepted as a poster because it introduces a novel framework, Con4m, for time-series classification and temporal action segmentation that leverages contextual information to enhance prediction accuracy. The proposed method creatively combines time-series classification and segmentation into a unified framework, which is valuable for dealing with complex sequential data with inconsistent labels. The originality of the work lies in using a Gaussian kernel to encode contextual prior information, which is more realistic than the assumption of independent and identically distributed time series segments commonly made by other models. The authors also conducted extensive experiments on various datasets, demonstrating the framework's effectiveness in improving coherence and robustness against label disturbances. While the results are promising, they also highlight several areas for future improvement.

However, the paper does have some limitations that should be addressed. Some parts of the paper lack clarity, such as the need for clearer explanations of specific terms and the overall architecture. Moreover, there are concerns regarding the model's scalability and efficiency, especially in large-scale applications where computational and memory resources could be a constraint. The evaluation setup could also be more comprehensive, including comparisons with models that specifically address change-point detection and segmentation, which would provide a more robust assessment of Con4m's capabilities. Despite these limitations, the paper presents a sound approach with a solid theoretical foundation and holds potential for future research and application, making it a suitable candidate for a poster presentation.